

# Collection efficiency of $\alpha$-pinene secondary organic aerosol particles explored via light scattering single particle aerosol mass spectrometry

Ellis Shipley Robinson[1,2], Timothy B. Onasch[3], Douglas Worsnop[3], and Neil M. Donahue[1,2]

[1]Carnegie Mellon University Center for Atmospheric Particle Studies, Pittsburgh, PA, USA
[2]Department of Chemical Engineering, Carnegie Mellon University, Pittsburgh, PA, USA
[3]Aerodyne Research, Inc., Billerica, MA, USA

*Correspondence to:* N.M. Donahue
(nmd@andrew.cmu.edu)

**Abstract.**

We investigated the collection efficiency and effective ionization efficiency for secondary organic aerosol (SOA) particles made from $\alpha$-pinene + O$_3$ using the single-particle capabilities of the Aerosol Mass Spectrometer (AMS). The mean count-based collection efficiency (CE$_p$) for SOA
across these experiments is 0.30 ($\pm$0.04 S.D.), ranging from 0.25 to 0.40. The mean mass-based collection efficiency (CE$_m$) is 0.49 ($\pm$0.07 S.D.). This sub-unit collection efficiency and delayed vaporization is attributable to particle bounce in the vaporization region. Using the coupled optical and chemical detection of the light scattering single-particle (LSSP) module of the AMS, we provide clear evidence that "delayed vaporization" is somewhat of a misnomer for these particles:
SOA particles that appear within the chopper window do not vaporize at a slow rate; rather, they flash-vaporize, but often not on the initial impact with the vaporizer, and instead upon a subsequent impact with a hot surface in the vaporization region. We also find that the effective ionization efficiency (defined as ions per particle, IPP) decreases with delayed arrival time. CE$_p$ is not a function of particle size (for the mobility diameter range investigated, 170-460 nm), but we did see a decrease in
CE$_p$ with thermodenuder temperature, implying that oxidation state and/or volatility can affect CE$_p$ for SOA.

## 1   Introduction

Organic aerosol (OA) comprises a significant fraction of submicron atmospheric particulate mass, ranging from 20-90% (Kanakidou et al., 2005; Jimenez et al., 2009). OA has been shown to have

negative impacts on human health (e.g. Lozano et al., 2013) and remains highly uncertain in its effect on radiative climate forcing (e.g. Solomon et al., 2007). The physical and chemical characteristics of OA can vary dramatically, and depend strongly on source, location, atmospheric age, and other factors. Despite the ubiquity and importance of OA, real-time measurements are technically chal-

lenging due to the wide range of chemical composition, particle size, and volatility represented by OA in the atmosphere.

The Aerosol Mass Spectrometer (AMS, Aerodyne Research, Inc.) is used widely in both ambient and laboratory measurements of OA. It has enabled significant advances in our understanding of how organic aerosols form (Craven et al., 2012), age (Aiken et al., 2008), and mix, (Robinson et al., 2013) by providing real-time measurements of size-resolved composition and mass for submicron,

non-refractory particulate matter (NR-PM$_1$). However, a lingering challenge with full quantification of NR-PM$_1$ in the AMS is the mass collection efficiency (Canagaratna et al., 2007), which is the ratio of the measured AMS mass signal to the actual NR-PM$_1$ mass concentration. Another factor influencing calibration of the AMS mass response is the species-specific relative ionization efficiency (RIE) of analyte; this is relative to a calibrant, typically ammonium nitrate. However,

this value is assumed to be roughly similar for all organic species (Jimenez et al., 2003), and is not subject to matrix effects. To obtain quantitative agreement between the AMS and other collocated instruments in field campaigns, a mass collection efficiency (CE$_m$) is usually applied to correct for the consistently lower AMS-measured mass. CE$_m$ can be written as:

$$CE_m = \frac{S_{AMS}}{S_O} \tag{1}$$

where $S_{AMS}$ is the signal from the AMS and $S_O$ is the signal of the another (perfectly calibrated) instrument. Importantly, this calculation assumes ideal operating conditions for both instruments and the application of all other appropriate correction factors. For example, Drewnick et al. (2003), in a sulfate aerosol intercomparison study, applied a scaling factor of 2.41 (CE$_m$ = 0.42) to the AMS-measured sulfate to achieve good agreement with a collocated particle into liquid sampler (PILS)

instrument. In another example, Middlebrook et al. (2012) recently reported parameterizations of CE$_m$ for ambient sulfate-containing particles that could be used to predict CE$_m$ based on particle acidity and mass fraction of ammonium nitrate. High CE$_m$ values were consistent with predicted liquid phase-state at 298 K. Thus, CE$_m$ should be thought of as a sensitivity factor that varies for particle types with different compositions and phase-states.

Huffman et al. (2005) define CE$_m$ as the product of three, size-dependent terms:

$$CE_m(d_{va}) = E_L(d_{va}) \times E_S(d_{va}) \times E_B(d_{va}) \tag{2}$$

where $E_L(d_{va})$ is the lens transmission efficiency as a function of aerodynamic diameter, $(d_{va})$; $E_S(d_{va})$ is the striking efficiency of particles on the AMS vaporizer transmitted through the lens to the time-of-flight chamber; and $E_B(d_{va})$ is the vaporization efficiency of particles that strike the vaporizer



surface, also known as "bounce" because particles can bounce away from the vaporizer surface and
escape detection. Any particle that enters the instrument, but is not detected by the mass spectrometer
due to any of these three loss terms contributes to the mass discrepancy between the AMS and
another (perfectly-calibrated) mass measurement.

Previous measurements and models have characterized the loss of particles in the lens region and
orifice, and have shown near-unit transmission efficiencies for particles in the size-range of 60-600
nm. However, for particle populations whose distribution is significantly outside of the transmission
window, especially for larger particles, $E_L$ can contribute significantly to $CE_m$ (Quinn et al., 2006).
The striking efficiency is a measurement of the divergence of the particle beam upon expansion into
the time-of-flight (ToF) chamber. While spherical particles can be narrowly focused on the 3.8 mm-
wide vaporizer over the distance of the ToF chamber, non-spherical particles can diverge from the
focused beam, causing sub-unit values of $E_S$ (Huffman et al., 2005). Studies show this term accounts
for very little particle loss for ambient aerosol (Salcedo et al., 2007), as well as laboratory SOA and
$(NH_4)_2SO_4$ (Docherty et al., 2013). $E_S$ can be well-characterized by the use of a beam-width probe
(Huffman et al., 2005).

In the majority of cases, the largest uncertainty and largest contributor to sub-unit $CE_m$ is the parti-
cle bounce term, $E_B$. Particle bounce has long been known to confound particle measurements, such
as impactors and surface-desorption mass spectrometers (e.g Myers and Fite, 1975; Virtanen et al.,
2010). A limited number of studies have investigated the nature and root cause of particle bounce
for laboratory aerosols in the AMS. Alfarra (2004) identified particle phase-state as a controller of
particle bounce for a selection of laboratory organics, where particle phase state was inferred from
the room temperature properties of the bulk materials. Liquid particles had near-unit $CE_m$, while
solid particles had much lower efficiencies ($CE_m$ = 0.2 to 0.5). Matthew et al. (2008) found phase
state to govern particle bounce as well. Ammonium nitrate particles, thought to be metastable liquids
at their experimental conditions (Lightstone et al., 2000), displayed high $CE_m$, while dry ammonium
sulfate particles had $CE_m$ = 0.22, which increased to 0.73 when the particles were hydrated and
deliquesced. Matthew et al. (2008) also found that $CE_m$ for dry ammonium sulfate increased as
the thickness of a liquid dioctyl sebacate coating layer increased. In some chamber experiments,
Bahreini et al. (2005) showed this same increase in $CE_m$ for dry ammonium sulfate particles with
condensation of an SOA layer. However, for other SOA experiments, $CE_m$ for ammonium sulfate
seed particles actually decreased with the condensation of SOA, implying that the SOA phase-state
was highly variable in their experiments and/or that other factors also govern particle bounce in the
AMS, such as compositon or volatility. Similarly, Robinson et al. (2015) showed that CE of liquid
squalane (CE ∼1) particles decreased following SOA condensation. Docherty et al. (2013) report
an inverse relationship between $CE_m$ for chamber-generated SOA and the $f_{44}/f_{57}$ ratio (where $m/z$
44 and $m/z$ 57 are comprised almost solely of signals from $CO_2^+$ and $C_4H_9^+$, and $f_i$ is the fraction of
($m/z_i$) to the total organic signal). This implicates oxidation state as either a factor influencing $CE_m$,



or as a proxy variable for what makes a particle bouncy.

An even smaller number of studies have used the light scattering single-particle (LSSP) module of the AMS to investigate collection efficiency, despite its ability to provide a real-time, particle number-based measurement of $E_B$. When $E_s$ and $E_L \sim 1$, collection efficiency is equal to the bounce efficiency ($CE \sim E_B$) We denote this number-based collection efficiency as $CE_p$ for "particle collection efficiency," which is defined as:

$$CE_p = \frac{\text{Particles with ion signal above threshold}}{\text{All particles}} \qquad (3)$$

Cross et al. (2007) first introduced LSSP as a method to resolve real-time densities of externally-mixed aerosols. Cross et al. (2009) later described the ability of LSSP to measure $CE_p$ for ambient particles from Mexico City, and found that a significant fraction of the optically-detected particles were either undetected by the mass spectrometer due to bounce (hereto referred to as "null") or exhibited signal at a time much later than would be expected based on their *in situ* measured velocity (referred to as "delayed"). "Prompt" particles, those that gave an appreciable chemical ion signal when they were expected to, made up only 23% of the measured aerosol, with the delayed fraction at 0.26 and the null fraction at 0.51. Liu et al. (2013) also report $CE_p$ for ambient measurements taken in Bakersfield, CA (Cal-Nex). They report a 0.46 prompt fraction, 0.06 delayed, and 0.48 null, and found a slight size-dependence in the campaign-average $CE_p$, which exhibited a maximum around $d_{va} = 600$ nm (0.52) and a minimum (0.42) for large particles. Slowik et al. (2009) compared $CE_m$ (density-corrected SMPS/AMS comparison) and $CE_p$ for an ambient biogenic SOA event, and found them to be equal.

Here we further explore the use of LSSP to identify the nature of collection efficiency for lab and chamber-generated aerosols. We quantify particle bounce for SOA from $\alpha$-pinene ozonolysis, as well as ammonium nitrate, ammonium sulfate, and squalane. We illustrate the difference between mass-based and number-based CE, which are not necessarily the same even for monodisperse aerosol, due both to decreasing effective ionization efficiencies for delayed particles (defined as ions per particle or IPP) and mass that registers at the detector on timescales much longer than the chopper cycle. We show that IPP decreases with delay time, that $CE_p$ is not a function of size for the SOA in this study, and that low volatility and/or high oxidation state decreases $CE_p$ for SOA.

## 2 Methods

### 2.1 Particle generation and sampling

We prepared inorganic aerosols (ammonium nitrate, ammonium sulfate) by atomizing dilute solution (1 g/L) using a constant output nebulizer (aerosol generator model 3076; TSI, Inc.). We sent these particles through a krypton neutralizer (10 mC), then size-selected them using a differential mobility


analyzer (DMA; classifier model 3080; TSI, Inc.) before sampling.

We sampled size-selected SOA in this same manner, but with a different preparation procedure. We injected a 1.2 $\mu$L aliquot of $\alpha$-pinene (Sigma Aldrich, >99%) into a clean and dry (RH< 3%) 100-L Tedlar sample bag (SKC, Inc.) at an estimated mixing ratio of ($\sim$ 2 ppm) and charged the bag

with excess ozone. This SOA formed at a high concentration ($C_{OA} \approx 1500\ \mu$g m$^{-3}$). This allowed us to study homogeneously nucleated SOA with the single-particle capability of the AMS, as the scattering laser requires large (d$_{va} \geq 180$ nm) particles. See Figure 1 for the general experimental schematic.

We produced squalane aerosols directly in the 12 m$^3$ Carnegie Mellon University smog chamber,

described elsewhere in greater detail (e.g. Robinson et al., 2015). In brief, we flushed the smog chamber continuously for >12 h with clean, dry air (cleaned with HEPA, silica-gel, and activated-carbon filters in series) to ensure low background particle, organic vapor, and water vapor concentrations. We prepared squalane particles by flash vaporization using a small, resistive stainless-steel heater. We placed a small aliquot of squalane (0.75 $\mu$L) on the heater surface, which we then inserted into

the smog chamber. Clean dispersion air flowed through the heater to carry and mix the squalane plume into the chamber while we power-cycled the heater for 10 minutes. Pure squalane particles formed as the vapor plume cooled.

We measured ensemble particle volume and number concentrations using a Scanning Mobility Particle Sizer (SMPS; TSI, Inc.). We measured ensemble composition and mass with the High-

Resolution Aerosol Mass Spectrometer (HR-Tof-AMS; Aerodyne, Inc,) operated in single-reflectron V-mode, fully described by DeCarlo et al. (2006). We acquired single-particle mass spectra using the light-scattering single-particle (LSSP) module coupled to the HR-ToF-AMS. We analyzed single-particle AMS data using Sparrow 1.04D[1], and ensemble AMS composition data using SQUIRREL 1.51[2].

## 2.2 Operation of light-scattering module

The LSSP module has been described in detail elsewhere in the literature (Cross et al., 2009). Briefly, the LSSP module consists of a continuous-wave laser (405 nm, 50-mW; LC BCL-050-405; Crysta-Laser) that crosses the collimated particle beam within the time-of-flight region of the AMS. Scattered light from sampled particles is collected by an ellipsoidal mirror that focuses the light onto

a photomultiplier tube. This light-scattering signal constrains the particle's velocity between the opening of the AMS chopper and the laser, allowing for the calculation of the vacuum aerodynamic diameter. It also prompts collection of individual mass spectra over the entire chopper cycle (e.g. 200 spectra/chopper), allowing for the identification of signals from individual particles within the

---

[1]Sparrow 1.04A,written by D. Sueper, Aerodyne Research Inc. and University of Colorado at Boulder; available at http://cires.colorado.edu/jimenez-group/ ToFAMSResources/ToFSoftware/index.htmlAnalysis4

[2]SQUIRREL 1.51,written by D. Sueper, Aerodyne Research Inc. and University of Colorado at Boulder; available at http://tinyurl.com/tofams?analysis





full chopper cycle. Saving at this data rate without the laser triggering (meaning all chopper cycles,
not just ones containing particles) is not practically useful, as it results in an unmanageable data
load. For example, when Drewnick et al. (2005) collected ToF-AMS single-particle data without
any triggering mechanism, of the 2.41 GB of data they collected, only 4 MB represented meaningful
single-particle spectra after applying their thresholding algorithm. The LSSP enables continuous
single-particle detection at a high duty cycle for the long timescales of chamber studies or ambient
sampling.

For data processing, we used an operationally-defined light-scattering threshold of five (signal-to-
noise, $S/N$) to identify particle events, and a mass threshold of six ions to identify a detected particle
to be further considered for particle classification, similar to Liu et al. (2013). For ammonium
sulfate, ammonium nitrate, and SOA, we used Sparrow's default ion list ($m/z$ 15, 30, 35, 36, 41, 43,
46, 48, 55, 57, 64, 71, 73, 80, 81, 98) for identifying particle events in the mass spectra of each
chopper cycle. We used a different list of deuterated ions ($m/z$ 48, 50, 66, 82, 98) to identify MS
events for $d_{62}$-squalane particles. We processed a subset of SOA experiments with an adjusted ion
list based on the 13 highest-signal ions for SOA that do not have significant background interferences
identified with MS mode spectra ($m/z$ 15, 26, 27, 29, 41, 42, 43, 44, 53, 55, 65, 67, 69, 79), but our
collection efficiency results were not sensitive to this change.

At the number concentrations of the high-$C_{OA}$ SOA experiments, coincident particles—multiple
particles sampled in a single chopper cycle—were present (13% of particles were coincident), but
identifiable from the scattered light signal. For typical smog chamber and ambient number concen-
trations (e.g. $\leq 2,000$ cm$^{-3}$), the probability for coincidence is rare. We expect $\sim 1$ particle per
chopper cycle for a 1% chopper slit ($\sim 70$ $\mu s$ wide) at typical conditions. We filtered out coinci-
dent particles (identified by multiple instances where the light scattering $S/N > 5$ during a single
chopper cycle) using the Sparrow analysis program and we did not consider them in our analysis or
calculation of CE$_p$.

### 2.3 Calculation of collection efficiency

We classified individual particle events based on how they interacted with the vaporizer, both in
terms of their effective ionization efficiency and vaporization quickness. As defined in Cross et al.
(2009), particles categorized as "prompt" arrive at the mass detector within a narrow time range after
they would be expected to arrive based on their measured velocity in the ToF region and assuming
instantaneous vaporization/ionization. The operationally-defined boundary between the prompt and
delayed particles is when the actual arrival of the mass signal differs from the expected arrival time
by 20% or more. In other words, we compared the measured arrival time at the detector (MS$_{arrival}$) and
the LS-estimated arrival time (LS$_{arrival}$) based on the measured velocity between the chopper and laser
to draw the boundary between prompt particles (MS$_{arrival}$/LS$_{arrival}$ $< 1.2$) and delayed particles (MS$_{arrival}$
/LS$_{arrival}$ $> 1.2$). As we shall show, this particular value for determining the boundary between prompt



and delayed particles is arbitrary.

LSSP provides an internal number-based measure of the AMS collection efficiency (Cross et al., 2009). The wide laser beam ($\approx 2$ mm), relative to the width of the particle beam ($\approx 0.5$ mm), allows for near complete optical detection of particles above the detection limit of the laser ($d_{va} > 180$ nm). The LSSP-based $CE_p$ is the comparison between the optically-detected particles (i.e. all particles

that enter the TOF region and that will hit the vaporizer surface) and the number of particles that are chemically-detected (i.e. give signal in the mass spectrometer). For all particles sampled here, $E_s$ and $E_L$ are reasonably assumed to be 1. Thus, in terms of the categories prompt, delayed, and null, the general definition of $CE_p$ from equation 3 can re-written as:

$$CE_p = \frac{N_{prompt} + N_{delayed}}{N_{prompt} + N_{delayed} + N_{null}} \qquad (4)$$

where e.g. $N_{prompt}$ is the number of prompt particles. In this formulation, we consider both prompt and delayed particles as those that give meaningful chemical signals at the detector, though it may be of interest in other studies to look at the $CE_p$ from e.g. only prompt particles. We are equating $CE_p$ with $E_B$, a reasonable assumption for the aerosols studied here as they all fall within the lens transmission window ($E_L = 1$) and are spherical (Zelenyuk et al. (2008)) and therefore do not exhibit

significant divergence from the particle beam ($E_s = 1$). However, it is important to note this collection efficiency accounts only for whether or not a particle was observed in the mass spectrometer, and does not account at all for signal strength above the detection threshold.

## 3  Results and Discussion

### 3.1  Delayed vaporization PToF artifact

It is standard practice to present comparisons between the mass-weighted size distribution from the SMPS and the particle time-of-flight mass distribution from the AMS to compute density and collection efficiency (DeCarlo et al., 2004; Kostenidou et al., 2007; Shilling et al., 2009). The SMPS size-distribution is multiplied by the density to align the mode diameters according to,

$$d_{va} = \frac{\rho_p}{\rho_0} \frac{d_{ve}}{\chi} \qquad (5)$$

where $\rho_p$ is particle density, $\rho_0$ is standard density (1 g cm$^{-3}$), and $\chi$ is the dynamic shape factor, which is equal to one for spherical particles and is assumed to be true in the case of SOA from $\alpha$-pinene ozonolysis (Zelenyuk et al., 2008). For spherical particles, $d_{ve}$, the volume equivalent diameter, is equal to mobility diameter.

For this example experiment, where 370 nm SOA particles were size-selected using a DMA,

shown in Figure 2, we estimate the density to be 1.1 g/cm$^3$ from aligning the AMS and SMPS mass distribution mode diameters. The shaded blue area is the frequency of optically-counted particles





as a function of size, as measured by light-scattering in the AMS. Like the SMPS distribution, this histogram is tight, as we expect it to be for size-selected particles. However, even after shifting the SMPS distribution by the density, the agreement between the SMPS- and AMS-derived mass distributions degrades considerably at large diameters.

We explore the nature of the divergence between the AMS PToF mass distribution and the SMPS-derived mass distribution at large apparent diameters using data from LSSP mode. We show the flight path, and resulting data, for a particle in the LSSP-AMS in Figure 3 (similar to Figure 7 in Cross et al., 2009). The scattered light pulse (magenta trace) triggers acquisition of mass spectra over the entire chopper cycle. Individual extractions from the mass spectrometer, which are usually averaged together over tens of seconds to minutes, are resolved at $\sim 30$ $\mu s$ (the ToF-MS pulser period) in single-particle mode (orange trace). Using the distance between the chopper and the point of intersection between the laser and particle beams, a flight velocity is calculated and used to predict the arrival of the particle's ions at the mass detector, assuming instantaneous vaporization and ionization. We show the mass signal as a function of time-of-flight for the chopper cycle in orange. For some particles, the arrival of the ions at the detector is significantly offset ("delayed") from the predicted arrival time. This offset (labeled "$\delta$" in Figure 3) is used to categorize particles into prompt and delayed categories, further discussed in Section 2.3.

Figure 4 shows total ion signals from individual particles (grey circles) along with total summed signals of prompt (blue) and delayed (red) particles as a function of time-of-flight. We see that the large-diameter PToF tail (green) matches the delayed particle distribution. Additionally, none of the prompt particles have measured times-of-flight greater than 4 ms. As described in Cross et al. (2009) for ambient OOA measured in Mexico City, the physical basis for the broadened PToF distribution at large diameters is particles with delayed vaporization, which comprise a significant fraction of the measured single particles in this SOA experiment (19% of all particles). However, the mechanism of the delayed vaporization has not yet been fully described for SOA from $\alpha$-pinene $+$ $O_3$.

### 3.2 Collection efficiency

The average $CE_p$ across all experiments was 0.30 ($\pm 0.04$), while the average $CE_m$ was 0.49 ($\pm 0.07$). We calculated $CE_m$ using equation 1, where $S_{AMS}$ is the AMS-measured mass from MS mode and $S_O$ is the density-corrected SMPS-measured mass. Like Cross et al. (2009), we see that $CE_m > CE_p$, which likely reflects two differences between the mass-based and particle-based collection efficiencies. First, by definition, null particles in LSSP mode, those which do not register mass above the 6 ion threshold, provide no chemical information. LSSP can only tell us that these particles bounced away from the vaporizer. However, there are examples (e.g. Huffman et al., 2009) where particulate mass is detected by the AMS on very long timescales (5 s) compared to the length of the chopper cycle window (5 ms). While a particle defined as "null" provides no chemical information whatsoever in LSSP mode, it is likely that not all null particles are created equal: some bounce away from the





vaporization/ionization region altogether and are not measured at all, while some bounce from the vaporizer cone but still do evaporate at very long timescales relative to the chopper cycle. Evidently,

the sum of some number of these particles from the null category do result in detectable mass on timescales longer than the chopper cycle, as evinced by $CE_m$ being significantly greater than $CE_p$.

Secondly, some particles that would register mass above the LSSP threshold may be delayed such that their mass signal registers at the detector just beyond the chopper cycle. As depicted in Figure 4, the delay times for some particles are just beyond the chopper cycle window that we used for

these experiments, as there are still mass signals arriving at the very right edge of the plot where the cycle ends. For aerosol types with a high delayed fraction like this SOA, a longer chopper-cycle would better accommodate these particles with long (2 ms) delay times. Thus, while LSSP provides an *in situ* measurement of the AMS collection efficiency, it is important to distinguish between the LSSP-based (eqn. 4) and mass-based (eqn. 1) calculations of collection efficiency.

**3.3 Delayed particle signal strengths**

Despite nearly equal numbers of prompt (17% of all particles) and delayed particles (19% of all particles) for this SOA, these two particle categories do not contribute equal mass signal to the detector. As shown in Figure 5, prompt particles produce significantly more signal per particle than delayed particles even though they are all the same size. We plot in Figure 5 a histogram of ions

per particle (IPP), normalized so that the sum of the bins for each category is one. This figure shows that the effective ionization efficiency for prompt particles is higher than that of delayed particles. Note that this "effective" ionization efficiency is not only a function of the ionization efficiency of the molecules being ionized by the 70 eV source (a molecular property), but also convolves the instrument sensitivity to particles that may be vaporized in a sub-optimal location

(for ion extraction). If delayed and prompt particles had the same IPP, the delayed vaporization tail in the AMS mass distribution for SOA shown in Figure 2 would be even more pronounced.

The single-particle mass signal (IPP) is a smooth function both within the prompt and delayed categories, possibly providing reason to redefine what it means to be "prompt" vs "delayed." Figure 6 shows a steady decrease in the average IPP as a function of delay time for delays shorter than 1 ms.

For delay times longer than 1 ms, the IPP is constant with delay time. The error bars represent the standard error of the mean within each bin, while the gray shadow shows the standard deviation for each bin reflecting the inherent spread of single-particle mass signals. For comparison, we include on the plot the average IPP value across all prompt and delayed particles, which is very similar to its calculated value based on the calibration ionization efficiency (IE) and the default relative

IE (RIE) value for organics of 1.4. It should be noted here that, as is done in most analysis of AMS data, converting from the nitrate-equivalent mass to the absolute mass measurement for a given non-refractory species (e.g. organics, sulfate, chloride, etc.) requires the application of both a species-specific values of CE as well as RIE (see e.g. equations 3.8 and 3.9 in Alfarra, 2004).



Thus, any measurement of CE also has inherent value into understanding RIE for a given species. Figure 6 illustrates this, as the measured average IPP for all particles matches the calculated value. However, clearly the least and most delayed particles have IPPs much different than the average, and thus particle bounce the associated loss of signal significantly affects IPP for a given particle. Measurements of RIE for various species using the AMS, as have been reported by e.g. Mensah et al. (2011) and Silva et al. (2008), is only possible when CE for the sampled aerosol particles is well-known. Given that LSSP measures CE inherently, easier and more routine measurements of species-specific RIE values, especially in ambient datasets, should be made possible with application of the LSSP module.

Plotting the accumulated particle counts as a function of delay time shows how single-particle information from LSSP mode can be used to best understand the response of the AMS to different particle types, each with its own sensitivity in the instrument (Figure 7). We scale the traces in Figure 7a by their measured $CE_p$ values (from equation 4). The effect of delay time on IPP is absent for ammonium nitrate, the standard mass calibrant for the AMS, because all particles arrive within the first few delay time bins. Squalane, a liquid at room temperatures with a near-unit $CE_p$, largely accumulates its signal at small delay times, but is noticeably slower to do so than ammonium nitrate. This is likely attributable both to the lower volatility of squalane and to the larger molecular weight of squalane (423 g/mol) compared to ammonium nitrate (80 g/mol). We estimate the squalane vapor pressure using SIMPOL (Pankow and Asher, 2008), and use the ammonium nitrate vapor pressure reported by Richardson and Hightower (1987)): Ammonium nitrate is more volatile than squalane ($\sim$ 30 $\mu g$ m$^{-3}$ and $\sim$ 0.1 $\mu g$ m$^{-3}$, respectively). Saleh et al. (2016) calculated the differences in evaporation timescales in the AMS vaporizer for species of different volatility, while Murphy (2015) discuss the molecular weight dependence on the movement of ions from the ion source to the ion optics region in a free molecular regime. Unlike both ammonium nitrate and squalane, however, SOA exhibits delayed vaporization and low $CE_p$, similar to crystalline ammonium sulfate, a possible indication of a solid or semi-solid phase state, extremely low-volatility material, or both.

Figure 7b shows how the total mass signal from single SOA particles accumulates faster than the particle counts as a function of delay time, as particles with low delay times contribute relatively more mass signal on average. The accumulation of single-particle counts is scaled by $CE_p$, while the single-particle mass accumulation trace is scaled by $CE_m$. We use $CE_m'$ to denote the mass collection efficiency calculated by comparing the AMS PToF vs SMPS mass, and $CE_m$ to denote the mass collection efficiency calculated according to equation 1. The difference between $CE_m'$ and $CE_m$ is the amount of mass that can be attributed particles counted as null by LSSP but are detected in MS mode at timescales much longer than the chopper cycle.





### 3.4 Nature of particle-vaporizer interactions

These results seem to indicate that when an aerosol type exhibits bounce, it also exhibits delayed
vaporization and thus lower effective ionization efficiency for some fraction of particles. In investi-
gating the offset between expected and actual arrival times, we tested two ideas about how the signal
at the mass detector would arrive for SOA within the LSSP chopper cycle. If an SOA particle strikes
and sticks to the vaporizer surface, but does not promptly vaporize, it should show an accumulation
of mass at the detector over time, beginning at the expected arrival time. It should sizzle. However,
if the particle bounces off the vaporizer without any significant evaporation, and somehow returns
to a hot surface at a later time, then the time-resolved arrival of ions should look similar to a prompt
particle that vaporizes upon impact, albeit after some time associated with its bouncy journey.

Indeed, when the mass arrival signals for an ensemble of single-particle events are averaged to-
gether, we see that prompt and delayed SOA particles have the same peak shape (Figure 8a). Here,
we display the average single-particle mass signal for particles with the same arrival time. We chose
two arrival-time bins with times-of-flight equal to 3.21 ms and 4.05 ms. All particles in each bin
are categorized as "prompt" and "delayed," respectively. The similar, sharp peak shape suggests that
delayed particles are truly delayed in starting their vaporization process, and not simply evaporating
at a slower rate. Drewnick et al. (2015) present the vaporization "event length" quantity, which is the
full width at half maximum (FWHM) of mass arrival signals from individual particles. In our study,
the time resolution of the mass arrival trace (determined by the pulser period, 30 $\mu$s) is on the same
order as the event length, which does not allow us to quantify the event length with any precision.
However, qualitatively we can say that prompt and delayed particles for the SOA presented here have
similar event lengths, and are on the order of $\sim$30 $\mu$s, similar to those measured by Drewnick et al.
(2015) for ammonium sulfate aerosol. We found nearly identical event lengths for prompt versus
delayed ammonium sulfate as well, indicating that ammonium sulfate exhibits the same behavior
of "flash vaporizing" even when the particles are delayed. The event length for ammonium nitrate
aerosol at low vaporizer temperatures, however, is fundamentally different (see Figure 8b); mass ar-
rives over a much longer timescale (1 ms), indicating that particles are sticking to the vaporizer and
slowly losing mass. Thus we conclude that delayed SOA, as well as ammonium sulfate, particles
must be bouncing around the ionization cage after initially striking the front of the vaporizer before
they finally land and flash-vaporize on one of the hot surfaces in the vaporization region (e.g. side
of the vaporizer, ionization cage). Our conclusion is the same as that of Cross et al. (2009), who
identified this mechanism acting on delayed particles in ambient measurements in Mexico City.

The AMS vaporizer is a cylindrical tube furnace ($r$ = 3.81 mm; $l$ = 20 mm) with a concave beveled
cone (60° included angle) serving as the stop for the particle beam. It is centered within an ionization
cage, a rectangular stainless steel housing ($h$ = 6 mm; $w$ = 8 mm; $l$ = 15 mm) which is open on each
end. The front end of the vaporizer is set back $\sim$10 mm from the front opening of the ion cage, and
$\sim$2 mm from the ion extraction volume, maximizing the intersection of the vaporized particle plume,



the electron beam from the filament, and the axis of ion extraction. Because of the long hot surface
of the vaporizer, which is housed inside a sheet-metal cage, this mechanistic picture of particles
bouncing around this region before finally landing on a hot surface is plausible. Importantly, for
this SOA, the actual vaporization of the particle still can be thought of as rapid—when the particle
finally does stick, it is vaporized and ionized on the same timescale as a "prompt" particle. Thus, the

"PToF broadening" shown in Figure 2 can be attributed to SOA particles bouncing around before
vaporizing, not slowly boiling off adsorbed material over time, as discussed in Salcedo et al. (2010)
for lead salts (e.g. $PbCl^+$), and in Drewnick et al. (2015) for sea salt and other semi-refractory
components (e.g. $ZnI_2$). Furthermore, this explanation is consistent with the decrease in IPP as
a function of delay time: when particles vaporize on e.g. the side of the vaporizer, they are in a

sub-optimal position for ionization of the resulting vapor plume and thus detection of the full single-
particle mass (Huffman et al., 2009). From Figure 6, the decrease in IPP with delay times up to 1
ms indicate an increasingly sub-optimal average vaporization location for the particle with respect
to the ionization region. For long delay times (>1 ms), the likelihood of the particle landing near
the ionization region becomes very low, but further delay does not influence the effective ionization

efficiency. As indicated by wide spread of IPP values for a given delay time in Figure 6, it is very
unlikely that a long-delayed particle can provide as many ions to the mass detector that the average
prompt particle can.

In Figure 6 we also show a secondary x axis of distance based on the nominal particle veloc-
ity. This is the distance traveled after the initial particle impact on the vaporizer, assuming elastic

scattering as the particle bounces. The inferred distance is long compared to the length scales of
the ionization region. We thus conclude that the particles are probably literally bouncing randomly
around the ionization region, impelled by surfaces that are rough at the length scale of the particles.
The top x-axis of Figure 6 shows our estimate for the nominal distance bounced for these 370 nm
particles. For this calculation we used the average measured velocity of the prompt particles, as mea-

sured between the chopper and laser. Comparing the length scales (∼1 cm) of the ionization cage
and vaporizer with our estimated distance bounced based on delay times, the most delayed particles
are experiencing many collisions with ionizer/vaporizer surfaces before finally vaporizing.

As a further check that the SOA particles measured in LSSP mode are rapidly vaporizing—just
simply doing so at a time later than would be expected based on their measured size and expected

time-of-flight—we increased the temperature of the vaporizer from 600 °C to 800 °C. Were the par-
ticles sitting on the vaporizer surface and slowly boiling, we would expect this temperature increase
to decrease the broadened PToF tail (Figure 9a). We do not see this effect (note: the degradation in
the organic PToF signal at 800 °C is due to low particle numbers at the end of our experiment due
to wall loss). However, when we coated SOA particles with squalane, a liquid at STP and a material

that exhibits essentially no particle bounce in the AMS ($CE_p \sim 1$), and the broadened tail of the SOA
mass distribution diminished. When we heated the chamber, causing the squalane to evaporate, the





broadened tail reappeared. This further supports this idea that delayed SOA particles are bouncing around the vaporizer-ionizer region before finally flash-vaporizing (Figure 9b).

On the other hand, the PToF distribution for ammonium nitrate can be broadened by decreasing

the vaporizer temperature from 600 °C to 200 °C. Figure 9b shows the mass distribution of $m/z$ 46 ($NO_2^+$) for both vaporizer temperatures. The increase in PToF arrival times (which translates to the broadened mass distribution) with decreased vaporizer temperature indicates that these particles do stick to the surface, and have a reduced mass flux at lower temperatures, thus spreading the signal arrival out over time-of-flight (Figure 9b). Docherty et al. (2015), operating their vaporizer

temperature on a programmed cycle between 200 and 600 °C, also see PToF broadening for nitrate in ambient data. Mass arrival signals from individual ammonium nitrate particles at these low vaporizer temperatures are much longer (event lengths ∼200 $\mu$s, consistent with those measured by Drewnick et al., 2015) than those shown for prompt and delayed SOA particles in Figure 8. There seem to be different mechanisms for particle delay both for different operating conditions of the AMS and for

different particle types.

Consistent with this proposed mechanism—that delayed SOA particles are bouncing around and vaporizing on surfaces away from the vaporizer cone—there are differences in mass spectra between prompt and delayed particles. Figure 10 shows the difference mass spectrum between prompt and delayed particles for both SOA and ammonium sulfate, both of which exhibit a high delayed fraction.

We created average mass spectra for prompt and delayed particles by summing the single-particle spectra for each category and dividing by the number of particles. We then normalized these average spectra by the sum of ions across all $m/z$, and the difference mass spectra is the normalized prompt MS minus normalized delayed MS. Error bars indicate the propagated standard error of the mean at each $m/z$.

Several fragments are more prominent in either the prompt or delayed mass spectra, colored by blue and red sticks, respectively. For instance, $m/z$ 43 (mostly $C_2H_3O^+$) is higher and $m/z$ 44 ($CO_2^+$) is lower for delayed SOA particles; the acidic fragments $m/z$ 81 ($HSO_3^+$) and 98 ($H_2SO_4^+$) are higher in the delayed MS for ammonium sulfate particles while $m/z$ 48 ($SO^+$) and 64 ($SO_2^+$) are higher in the prompt MS for ammonium sulfate. The ammonium ion ($NH_4^+$) is enhanced in

the prompt MS, while ammonia ($NH_3^+$) is enhanced in the delayed MS for ammonium sulfate particles. We attribute these differences in mass spectra between prompt and delayed particles to the wide range of possible temperatures experieneced by delayed particles that have bounced away from the center of the AMS vaporizer. The lower temperatures at these sub-optimal vaporization positions (e.g. side of the vaporizer, on the ion cage, etc.) can lead to different fragmentation pathways, which

could be important for interpreting ambient single-particle spectra.

To support this hypothesis, we look at previous work conducted by Docherty et al. (2015). They show that acidic fragments from ambient ammonium sulfate measured during the Study of Organic Aerosols at Riverside (SOAR-2005) are enhanced when they lower the AMS vaporizer temperature





from 600 °C to 200 °C, which is consistent with our hypothesis that delayed ammonium sulfate
particles were vaporizing on cooler surfaces. Docherty et al. (2015) also show that ambient OA in
SOAR-2005 appeared more oxidized at lower vaporizer temperatures, as indicated by increased $f_{44}$
and increased O:C. While $f_{44}$ is slightly higher in our prompt SOA MS, perhaps indicating that the
prompt particles appear more oxidized, the rest of the mass spectrum shows that the delayed particles
are enhanced in oxidized fragments, while the prompt particles are enhanced in reduced fragments.
We see an enhancement in the delayed MS of $C_xH_yO$ fragments, such $m/z$ 71 ($C_4H_7O^+$), $m/z$ 83
($C_5H_7O^+$), and $m/z$ 97 ($C_6H_9O^+$). Other studies have found that $f_{44}$ does not change or even
decreases with lower vaporizer temperatures compared to the standard 600 °C; for example, Cana-
garatna et al. (2015) showed that $f_{44}$ decreases in the MS of cis-pinonic acid at 200 °C compared
to the standard temperature. Thus, the enhancement of these $C_xH_yO^+$ fragments in the delayed MS
is a more robust indicator than $f_{44}$ that our delayed SOA particles appear more oxidized than the
prompt ones. Excluding $f_{44}$, our data are consistent with Docherty et al. (2015) and the hypothesis
that our delayed particles are bouncing around the vaporization/ionization region before landing on
cooler surfaces and finally evaporating. Importantly, these data show that particles delayed due to
particle bounce, like ammonium sulfate and the SOA studied here, can have differences in their mass
spectra that need to be considered when analyzing ambient single-particle data.

### 3.5 Collection efficiency as a function of size and thermodenuder temperature

As reported previously in the literature, some studies have shown collection efficiency for OA to
be size (Liu et al. (2013)) and composition-dependent (Docherty et al., 2013). To investigate any
size-dependent collection efficiency that our SOA might have, we selected particles at different
mobility diameters with a DMA upstream of the AMS. Figure 11a shows $CE_p$ as a function of
selected mobility diameter. LSSP can also provide a size-resolved $CE_p$ for polydisperse aerosol (as
in Liu et al., 2013), as each optically-counted particle has an estimated $d_{va}$. Figure 11a also shows
$CE_p$ for polydisperse SOA from multiple smog-chamber experiments, which agree well with the
size-selected data. The $CE_p$ for SOA studied here was not a strong function of size between in
diameter range 170-460 nm. The mean $CE_p$ across all experiments for SOA was 0.3 ($\pm$0.04 S.D.),
and ranged from as low as 0.25 to as high as 0.4.

While $CE_p$ for this SOA is independent of size, we do observe a decreasing trend in $CE_p$ by pass-
ing the SOA through a thermodenuder. We sampled SOA alternately through a thermally-denuded
line, or through a bypass line of the same length held at the same temperature as the chamber. Figure
11 shows $CE_p$ plotted against thermodenuder temperature for an experiment where SOA particles
passed through a thermal denuder operating on a temperature ramp profile. The ramp program in-
creased temperature linearly over one hour from 27 °C to 130 °C, soaked at 130 °C for 10 minutes,
and then returned back to 27 °C at the same rate. Above 110 °C, almost all SOA evaporated in the
thermodenuder, making the $CE_p$ measurement impossible. The $CE_p$ values in Figure 11 are calcu-





lated for particles with 200 nm > $d_{va}$ >300nm to isolate the effects of volatility and/or oxidation
state on $CE_p$, instead of measuring smaller particles less likely to provide enough detectable mass
above the threshold.

We use temperature as a proxy variable for the volatility of the aerosol, because SOA particles
that have passed through the denuder will have had some fraction of their more-volatile components

removed, the amount of which increases with increasing temperature. We color data points in Figure
11 by $f_{44}$ as measured from MS mode bulk mass spectra, which is used in AMS analysis as both
a direct measurement of oxidation state and a proxy for OA volatility (Ng et al., 2011). These
data show that $CE_p$ is inversely related to either the SOA oxidation state, volatility, or both. These
results are consistent with the trend shown by Docherty et al. (2013), who saw decreasing $CE_m$ with

increasing oxidation state, though are within the range of scatter shown in Figure 11a for all SOA
experiments. It should be noted that this SOA is similarly oxidized ($f_{44}/f_{57} \approx 6$) as the least oxidized
SOA from their study ($f_{44}/f_{57} \geq 5$), which had a $CE_m$ of ~0.2 ($f_{44}/f_{57} \geq 5$). SOA sampled through
the bypass line during this same time period did not have any decrease in $CE_p$. It is not possible to
determine whether the decrease in $CE_p$ is attributable to changes in volatility or oxidation state, as

the two are coupled in our measurements. However, this example shows that LSSP can be used to
verify whether this trend in $CE_p$ with these compositional changes exists for other types of NR-PM$_1$.

## 4   Conclusions

In this study, we present LSSP AMS data that gives further insights into the nature of collection
efficiency for the common laboratory system of SOA from $\alpha$ pinene + O$_3$. SOA generated in these

experiments exhibited an artificial tail in the PToF distribution at large diameters, which we show
to be an artifact of delayed vaporization. However, by studying the arrival of mass signals for these
delayed SOA particles, we see that the signals can not be attributed to adsorption on the AMS vapor-
izer followed by slow evaporation. Rather, particles bounce off the vaporizer after primary impact
and vaporize on some subsequent impact with a hot surface in the vaporization region. This causes

the mass arrival at the detector to be delayed relative to the estimated speed from optical detection,
but is fundamentally different than slow evaporation from the vaporizer surface. A significant frac-
tion of SOA and ammonium sulfate exhibited this type of delayed vaporization, while ammonium
nitrate and squalane exhibited none. For delayed particles, the measured per-particle mass signal is
reduced, which we report as ions per particle as a function of delay time. The artificially broadened

PToF distributions would be even more prominent if the delayed particles had the same effective ion-
ization efficiency as prompt particles. However, some of the SOA particles counted as null evidently
evaporate on very long timescales relative to the chopper cycle, as indicated by $CE_m$ >$CE_p$. These
particles register no mass signal in LSSP mode, and are labeled as "null," though what fraction of
null particles will result in detectable mass on the long timescale of MS mode is not discernible using





these data. The reduced number of ions per particle of delayed particles means that the AMS PToF
signal for polydisperse distributions will be dominated by prompt particles, because larger prompt
particles with high IPP will overwhelm smaller delayed particles with lower mass and few ions per
unit mass. However, the large diameter tail in the AMS PToF spectrum should be regarded with
caution. Additionally, we use the LSSP to show that particle collection efficiency is not a function of

size for the size range explored ($170 < d_m < 460$ nm), but is related to the oxidation state/volatility
of this SOA.

Rather than being viewed as a limitation, collection efficiency should be viewed as a sensitivity
within the AMS that simply needs to be understood for a given system and that may provide addi-
tional useful information. We demonstrate here that using the LSSP capabilities of the AMS allows

users to gain further insight into a given aerosol system. Further work should be conducted to better
understand any compositional artifacts that may be attributable to delayed vaporization. Data of this
kind may also possibly be used for design improvements to the vaporization region.

*Acknowledgements.* This research was supported by grant CHE1412309 from the National Science Foundation.
The High-Resolution Aerosol Mass Spectrometer was purchased with Major Research Instrumentation funds

from grant CBET0922643 through the National Science Foundation, and generous support of the Wallace
Research Foundation. The authors would like to thank Rawad Saleh, Manjula Canagaratna, John Jayne, and
Eben Cross for useful discussions regarding data analysis. Thanks for John-Charles Baucom for his help with
the design and construction of the vaporizer used to prepare squalane aerosol.





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





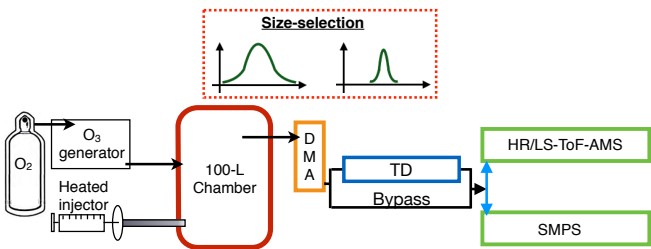

Fig. 1: Experimental setup for SOA CE experiments.

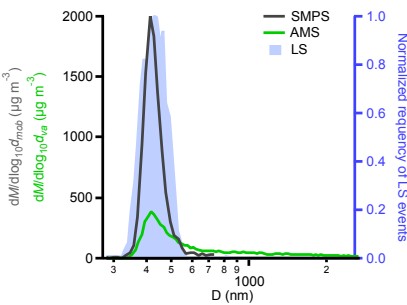

Fig. 2: Ensemble mass distributions from SMPS (black trace, adjusted for density) and AMS (green trace) of size-selected $\alpha$-pinene derived SOA particles with 370 nm mobility diameter for an example SOA experiment. Frequency of optically-counted particles (from LSSP) as a function of size shown in blue. For this instance, $CE_m$ = 0.39. The blue trace is normalized to 1 and plotted on the right axis as to have the same height as the SMPS trace, reflecting that optical detection in the AMS flight chamber is not affected by particle bounce.





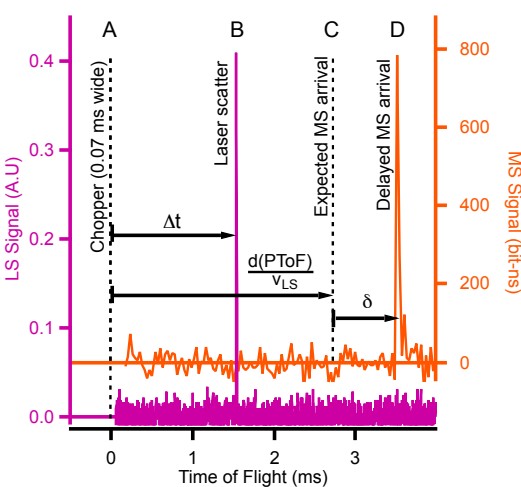

Fig. 3: Scattered light signal (magenta) and mass spectrometer signal (orange) as a function of particle time-of-flightfor an example delayed particle. The particle velocity ($v_{LS}$) is calculated by the measured time between the start of the chopper cycle (point "A") and detection of the scattered light peak (B). The velocity is used to estimate an expected arrival time of the chemical ion signal at the mass spectrometer (C) assuming prompt evaporation and ionization of the particle at the vaporizer. The difference between the expected (C) and actual (D) arrival times is denoted by $\delta$, and allows for the operational definition of prompt and delayed particle events.





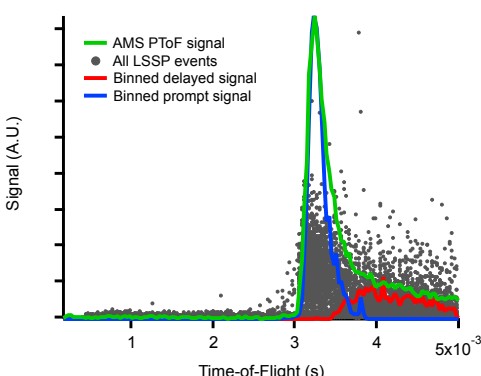

Fig. 4: Particle signal versus particle time-of-flight from the chopper to the mass detector are shown for ensemble mode (green trace) and for all detected single particle events (gray circles) in a representative SOA experiment (DMA size-selecting SOA particles with mobility diameter = 370 nm). Particles are sorted into either prompt or delayed categories based on their delay time. The mass signals for individual particles within each category are binned by flight time and summed to create the prompt (blue trace) and delayed (red trace) distributions.

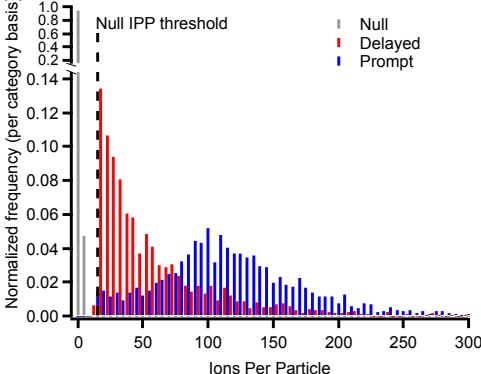

Fig. 5: Histogram of ions per particle for null (gray), delayed (red), and prompt (blue) particle categories for 370 nm SOA particles. The y-axis is the frequency of single-particle events within each category (in other words, all data for each category sum to one). The histogram bars for the delayed category are offset (by 5 ions) on the x-axis for clarity.





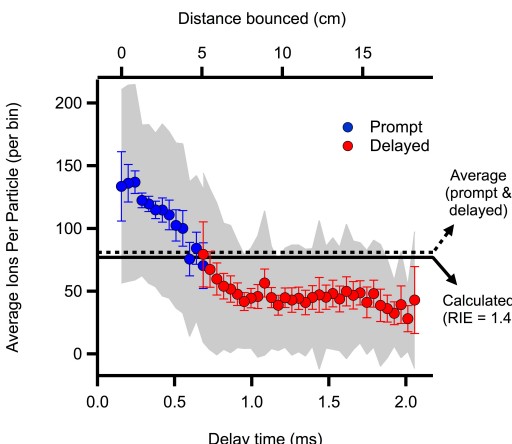

Fig. 6: Ions per particle as a function of the delay time between the expected time of arrival for the chemical ion signal and the actual time of arrival for 370 nm SOA particles. The data points are the arithmetic mean IPP value for a given delay time bin. Error bars are the standard error of the mean for each bin, which represent the precision of the average IPP values. The gray shadow behind is the standard deviation of ions per particle within each bin, which reflects the inherent spread of single particle signals at a given delay time for monodisperse SOA. Dotted line shows the average IPP for the entire ensemble, while the solid line shows the calculated IPP based on an ionization efficiency (IE) of 5e-7 and an RIE of 1.4 for organics compared to ammonium nitrate. We also estimate the nominal distance bounced (top x-axis) for these particles, assuming the average velocity of the size-selected particles measured between the chopper and laser.



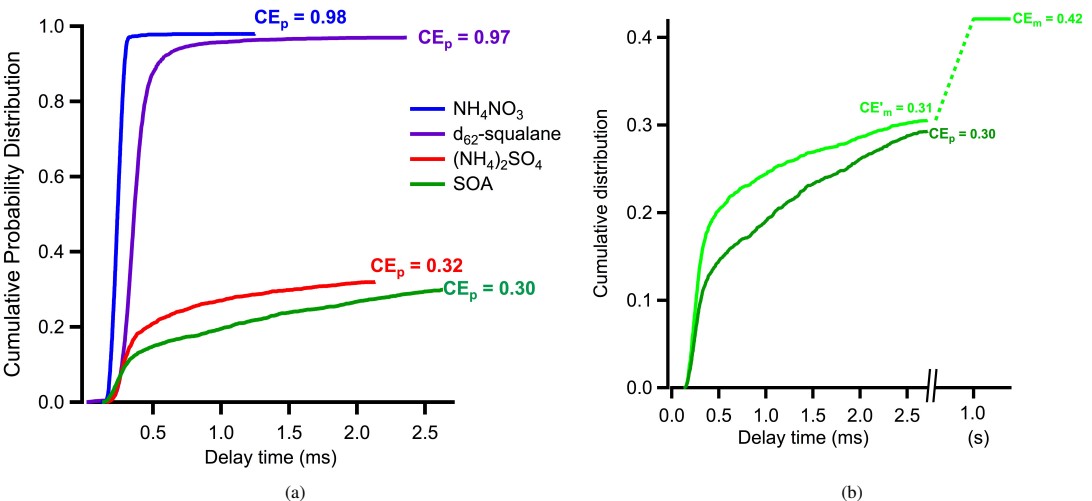

(a)  (b)

Fig. 7: (a) Cumulative probability distributions of particle counts as a function of delay time for ammonium nitrate (blue), $D_{62}$-squalane (purple), ammonium sulfate (red), and SOA (green). All traces are normalized by the respective $CE_p$ values, which is the average value across all experiments for that particle type. (b) Cumulative probability distributions for single particle counts (dark green) and single-particle mass (light green) for an individual SOA experiment. The dark green trace is scaled by $CE_p$. The light green trace, up to 2.5 ms delay time, is scaled by the mass collection efficiency as determined by comparing the AMS PToF-determined mass to the SMPS mass, according to equation 1. The broken axis represents additional mass seen beyond the window of the chopper cycle, and that mass is scaled according to the mass collection efficiency determined by the AMS mass seen in MS mode compared to the SMPS mass.





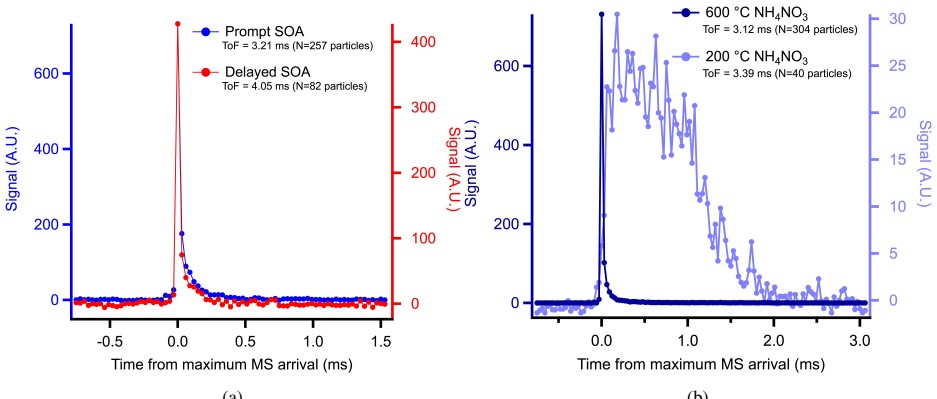

Fig. 8: Profiles of single-particle mass arrival for SOA (a) and ammonium nitrate particles (b) under different vaporization scenarios. (a) Plot shows total chemical ion detection as a function of time from arrival of maximum signal for SOA. The traces represent the average signal for all particles with the same MS arrival time. The two arrival time bins shown correspond to either all prompt (ToF bin = 3.21 ms, blue trace) or delayed (ToF bin = 4.05 ms, red trace) particles. N is the number of particles used to make the average trace. (b) Similarly, average chemical signals as a function of arrival time are shown for ammonium nitrate particles at two different vaporizer temperatures. The arrival of mass at the detector (event length) is significantly longer for ammonium nitrate at 200 °C compared to 600 °C





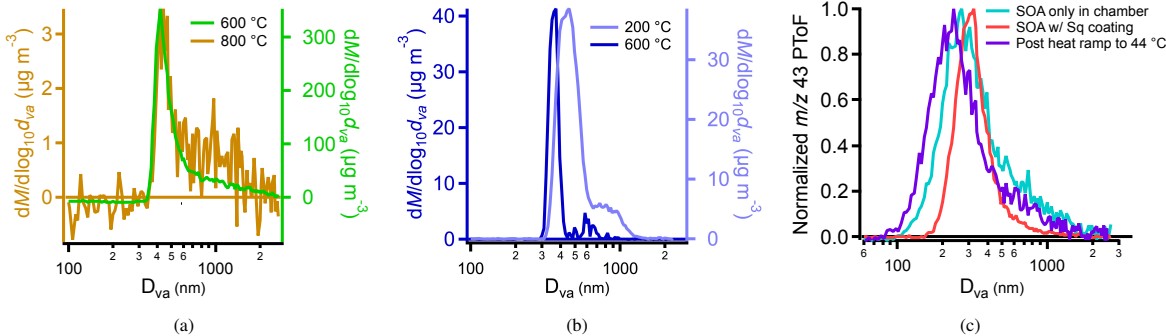

Fig. 9: Ensemble mass distributions of different particle types. (a) Organic mass distributions for $\alpha$-pinene derived SOA particles at two different vaporizer temperatures with DMA-selected mobility diameter of 370 nm: 600 °C (green) and 800 °C (brown). Note: the degraded signal at 800 °C is due to low-particle numbers due to wall-loss, as these data were taken at the end of an experiment where particle number was relatively low. (b) $m/z$ 46 PToF mass distributions for DMA-selected mobility diameter of 300 nm at the standard vaporizer temperature (600 °C, dark blue), and low temperature (200 °C, sky blue). (c) $m/z$ 43 mass distributions from SOA particles at three stages of a mixed-particle experiment: homogeneously-nucleated SOA (teal), SOA particles coated with squalane (red), and SOA/squalane particles after an increase in chamber temperature (purple). Note the disappearance of the delayed tail with the condensation of squalane, and the reappearance of the tail with heating despite the decrease in mode diameter.





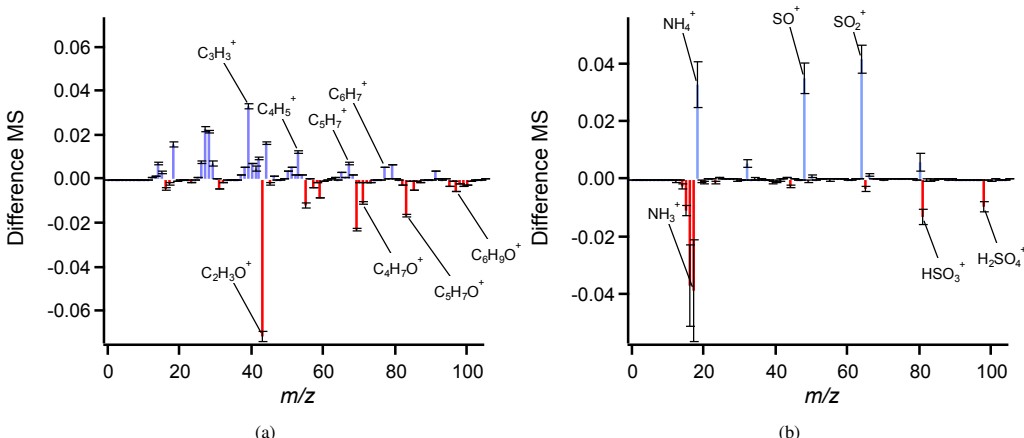

Fig. 10: Difference plots between prompt and delayed average mass spectra for (a) SOA and (B) ammonium sulfate. Plot is prepared by first normalizing each spectra by the total signal, and then subtracting the normalized delayed MS from the normalized prompt MS. Thus, mass fragments with positive values (blue) are enriched in the prompt MS, and those with negative values (red) are enriched in the delayed MS. Error bars are the propagated standard errors of the mean for each population.



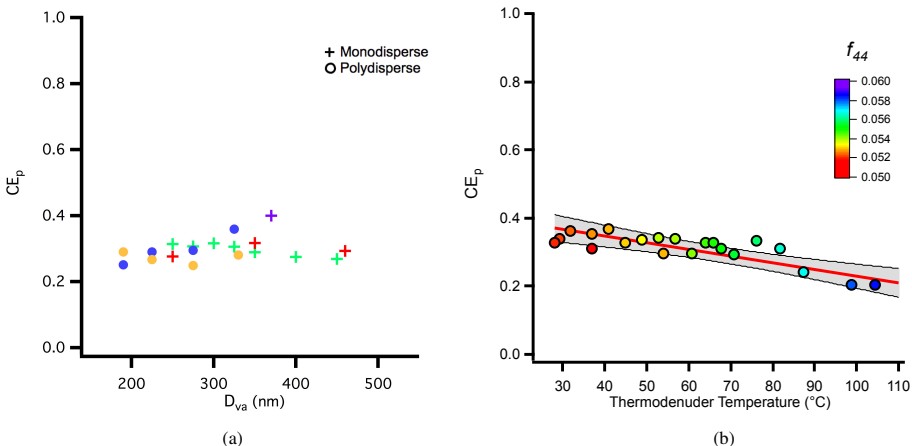

Fig. 11: (a) Particle collection efficiency as a function of $D_{va}$ for all SOA experiments. Data are from both size-selected experiments (crosses) and polydisperse SOA from a smog chamber (circles), with each color representing a separate experiment. (b) Particle collection efficiency for 370 nm size-selected particles (colored markers) as a function of thermodenuder temperature for a single SOA from $\alpha$-pinene experiment, colored by the fraction of *m/z* 44 ($f_{44}$) to the total organic mass measured in MS mode. Confidence intervals (95% CI) for a linear fit are shown (slope: -0.0020 $°\mathrm{C}^{-1}$).