# Peer review of "Collection efficiency of $\alpha$ -pinene secondary organic aerosol particles explored via light scattering single particle aerosol mass spectrometry"

_Atmospheric Measurement Techniques, 2016_

## Referee Comment (RC1) · Anonymous Referee #1 · 26 Sep 2016

This paper reports a detailed investigation of the AMS collection efficiency of organic particles made in a laboratory using a light scattering module. The results are not wildly inconisistent with previous results but there is a need for careful and systematic studies of this nature as we begin to understand some of the detailed internal processes of the AMS as we try to move on from simply dealing with these issues emprically. There is currently some debate on the exact nature of some of the empirical factors and how best to extrapolate these in a general manner, so papers of this nature are important. Furthermore, the paper lays down a framework that other studies could easily follow, which should allow for the better determination of accurate CEs and RIEs of organic

matter in the future.

The paper is well within AMT's scope (given how widely used the AMS and ACSM are) and was extremely well written. Asides from a few technical niggles, I have no hesitation recommending publication.

Technical comments:

Line 125: Bequerel (Bq) is the SI unit for activity.

Line 339: Please explain what you mean by 'sizzle'.

Figure 3: Please use two more distinct colours.

Figure 5: Rather than normalised frequency per bin, it would be better to do this as a probability density function (taking bin width into account). The data could then be legitimately plotted as lines (rather than sticks), making the figure easier to read.
* * *

---

## Referee Comment (RC2) · Anonymous Referee #2 · 18 Nov 2016

Review of Robinson et al

The paper by Robinson et al. presents a study of AMS collection efficiency for laboratory SOA from the commonly studied a-pinene + O3 system. It uses two techniques to quantify CE, the light scattering single particle mode of the AMS (LSSP, yielding CEp) and a comparison to an SMPS (CEm). It goes into an unusual level of detail in describing the results, and will represent a very useful addition to the AMS literature. I support publication in AMT, after the issues listed below are addressed.

Major issues

[Figure]

(1) A key problem is that the manuscript, as currently written, will be very confusing for most AMS users. The wording is quite specific and a clear connection to the way most users utilize the AMS is not given anywhere. My estimate is that 99% of published AMS figures use MS-mode data only, ∼1% use PToF mode, and ∼0.1% or less use LSSP mode. This may become even more skewed towards MS-mode data in the future as the number of ACSMs (that only have MS mode) surpasses and then greatly exceeds the number of AMSs in use. Thus it seems critical to clarify the implications explicitly. This could be done with a section such as "Implications for AMS and ACSM" users before the conclusions, or maybe a paragraph in the conclusions.

In particular the fact that some particles do not evaporate within the PToF timescale or don't give LSSP signals is of no direct relevance to users of MS mode data in terms of data processing etc. (even if they are still useful for users wishing for a deeper understanding of the instrument, but which lately are few and far between). Only the fractions that are not detected using MS mode (i.e. CEm) are relevant to the vast majority of AMS/ACSM practitioners.

As an example of this issue, in P8 L259-261 the wording is quite confusing. 5 s is not an arbitrary long timescale. It is the timescale of the MS mode of the AMS, from which mass concentrations are always reported. This should be clarified.

(2) There is a small fraction of the signal for non-refractory species that is detected over even longer timescales than the MS mode open / closed alternation, and thus contributes to slower variations in the AMS background signals ("MS closed"), as documented in e.g. Huffman et al. (ACP 2009, cited in the manuscript). For semi-refractory species, such very slow signals can be larger and even dominant, but even then they can be used to extract particle concentrations (see e.g. Salcedo et al. 2010).

Did the authors examine the variations in the background of the mass spectrometer (MS closed) during their experiments? It most likely contains information that is very relevant to this paper, in particular signals from the evaporation of bounced particles.

This should be analyzed and discussed in the revised paper.

(3) An important limitation of the results of this study is that the SOA was formed at extremely high concentrations (1500 ug/m3). Those concentrations are not atmospherically-relevant, and the composition of the SOA formed cannot be expected to be the same as for much lower, atmospherically-relevant concentrations. For example typical concentrations at Hyytiala, Finland (a location where a-pinene SOA is important) are ∼1 ug/m3, a factor of 1000 smaller than in this study. Shilling et al. (2009) showed that the composition of a-pinene + O3 SOA changed very strongly between 0.5 and 140 ug m-3, with very large changes below 30 ug m-3. The results of this study, while important in terms of characterizing the AMS detection system, should not be recommended for application for ambient particles in terpene-dominated environments. The fact that the SOA was generated at very high concentrations, and thus that it is not clear whether the results will be applicable to ambient SOA from this precursor and oxidant, needs to be stated prominently in the abstract and conclusions of the paper.

(4) The need to use extremely high concentrations would appear to be a critical limitation of this technique. It is important that the author discuss what the lowest concentrations that could possibly be studied with the LSSP method are, as well as possible methods to allow performing similar experiments at lower SOA concentrations. Did the authors try to use small seeds? E.g. using 10 nm seeds from (for example) an electrospray aerosol source), that would then be coated by a much larger amount of SOA. Presumably the CE would be dominated by the SOA coating, and insensitive to the initial seed. The authors are creative experimentalists and may have better suggestions, and it would be useful to document those in the paper.

(5) Fig. 5 and P9 L271. Apparently a very short chopper time (5 ms) was used for these experiments. This is unfortunate, as the AMS can typically be operated at >10 ms, which would have provided better data. Similarly the mass spec. pulsing frequency was too long (30 us), and a shorter frequency would have allowed recording single particle

events in more detail. Both items should be recommended in this paper to future experimenters wishing to use this technique, so that improved data can be acquired.

(6) P10, L301-307. I would argue that this is an extremely important point. I.e. the confusion of many users about the potential variability in CE, and the large effort needed to estimate it accurately, has obscured any trends in RIE that might be present on the laboratory and field data. Methods such as in the current paper where CE is quantified allow investigating RIE, which is a timely topic. It seems to me that this should be mentioned more prominently, e.g. in the abstract and conclusions.

(7) P11, L336-342. An alternative hypothesis (that appears much more likely than the 2 mentioned here) is that a particle bounces from the vaporizer without any evaporation, and then lands into a colder surface in the ionizer. The AMS ionizer cage is heated to ∼250C by the electron emission filament, and thus NO prompt evaporation at all should be expected for most species when impacting that surface. Given the geometry of the AMS ionizer, most bounced particles will impact the AMS ionizer cage. If they exit the cage region, they will end up on surfaces which are much farther and much colder (∼30C). This hypothesis can explain why substantial more particle mass is detected in MS mode than in PToF mode, as the colder temperatures (∼250C) result in far slower evaporation. Can the authors incorporate this hypothesis into their discussion, even if these particles may not be detected at all in PToF mode?

(8) P11, L344, this result can be influenced by "threshold bias". I.e. some particles may start to evaporate more slowly per the previous comment, but not be detected at all in by the ion threshold used in single particle mode. This appears obvious from the distribution of signals of the delayed particles in Fig. 5, which would have kept increasing in frequency towards lower signals than the threshold used here.

Maybe only those particles that impact the ion cage very close to the filaments, in an area that is likely much hotter than 250C, may lead to the sharp delayed particles.

Overall while this discussion is valuable, it is incomplete. A physical schematic (cartoon) of the different possibilities for particle trajectories and associated signals would be useful for readers of the papers and for future discussions (e.g. at AMS Users Meetings and in future papers). This should include particles completely missed due to threshold bias under slow evaporation. I strongly encourage the authors to add it here, and include the additional possibilities mentioned in the previous paragraphs (even if the present data do not allow characterizing all of them). Similarly, on P11, L375: this hypothesis is not proven with the current data, since other particles may have been completely missed due to threshold bias.

(9) P12, L388, and L395-397. This text and axis label should be "upper limit of distance bounced" as particles will lose energy and reduce their speed upon collision. The conclusion on L397 about many bounces being common is not really supported. E.g. if the particles lose 90% of their speed upon the first impact, then they may just evaporate on the 2nd impact, with a distance that the authors would calculate as 10 cm, but would only really be 1 cm. There is some data on velocity changes upon impact for much larger particles (e.g. Li et al., 1999). I did not find results for submicron particles on a quick search, but they are probably out there. One would expect that submicron particles will lose a much larger fraction of their speed, due to the far larger ratio of adhesion forces to inertia, compared to the data in Li et al. The authors should do a more thorough search and use the results to inform this discussion.

(10) P12 L401-402: this conclusion could be made if adhesion/adsorption forces did not change between the two temperatures. But for example Fig. 11 of Hu et al. (2016) shows that chloride detection in the standard vaporizer becomes slower when going from 600 to 835 C. This effect is not observed for other species, indicating that it depends on the specific chemistry of the system being studied. This seems to be the case here as well, see next comment. So I don't think that the conclusion about where this tail comes from is as obvious as the authors state. Note that the ionizer surfaces will be somewhat hotter with a hotter vaporizer, due to radiation heat transfer which is efficient at those temperatures.

(11) P12 L403-405. This statement is incorrect. It is clear from the data in Fig. 9a (of this paper) that SOA detection has a LONGER tail at 800C. Although the signal is noisy, a fit or smoothing of the data (as I did just by eye) clearly reveals this feature. Also only the front baseline region should be used to provide a zero level for the data ("DC marker region"), which will also increase the difference. At present the average of the front region is clearly negative.

(12) P12 L404-406, here it is unclear whether the signal plotted in Fig 9c is exclusive to a-pinene SOA. If the m/z 43 signal arises from both squalane and SOA (as expected from the NIST spectrum of squalene http://webbook.nist.gov/cgi/inchi?ID=C111013&Mask=200#Mass-Spec, and noting that the AMS spectrum will have substantially higher m/z 43 than the NIST one), it is possible that the signal shown in this figure is dominated by squalane, and thus that the conclusion reached here is incorrect. The authors have experience identifying signals dominated by one or another species/sources in binary mixtures, and that method should be applied to improve this graph and hopefully clarify whether the conclusion is correct.

(13) P13 L 409-420. This conclusion can only be made if the arrival times (initial onset of the signal) are shown for $NH_4NO_3$, which is not currently done. Fig 9b seems to show a secondary mode that could be due to delayed particles. The author should investigate and document this.

(14) P13 L 434-436 and Fig 10: the $NH_4+$ fragment is shown for ammonium sulfate. This is typically not detected in AMS spectra (e.g. Table 1 of Hu et al., 2016, which is derived from high-resolution data). Presumably the authors are analyzing unit-resolution data and mislabeled the ion. This fragment is almost certainly $HÂň2O+$, a major ion observed for $(NH4)2SO4$. More H2O is expected to arise arise from the decomposition of $H2SO4$, a process more important for the prompt particles, as Fig. 10b shows. It is important to fix this as it could otherwise cause a lot of confusion in the AMS community. (Alternatively, if the authors had high-resolution evidence of $NH_4+$

detection at such a large fraction as shown in the figure, it would be big news).

(15) P14 L 466-467: How can LSSP separate prompt and delayed particles of different sizes, when sampling a polydisperse aerosol? It seems (from e.g. Fig. 4) that this would be extremely difficult, unless all particles were prompt, which is not the case here. This is important since the polydisperse data in the blue experiment seem to show an increase with size, while the (much more trustable, in my opinion) monodisperse data do not.

(16) P2 L35: Jimenez et al does not state that RIE_org is assumed to be the same for all organic species. It does actually provide some cross-section data that could allow grouping hydrocarbons and oxygenates separately. The assumption of the same RIE_org for all organics in the field was introduced later based on experimental results at Aerodyne, and to my knowledge Canagaratna et al. (Mass Spec. Rev. 2007) may be the first publication of that assumption. To my knowledge it has never been published that all organics in the lab should be analyzed with RIE = 1.4, rather than calibrated for each experiment (especially given wide variability in CE of organics, as documented by e.g. Bahreini et al. (2005) and Docherty et al. (2013). Although that has been a common misinterpretation. See Jimenez et al. (2016) for further discussion of this topic.

(17) P16 L522-524. Returning to issue #1 and the implications for most AMS/ACSM users. I would argue that for the vast majority of AMS users, CE IS a serious limitation. While the information obtained is undoubtedly of high interest, very few AMS groups have the technical skill, equipment, and time to perform and interpret analyses such as the ones in this paper (as evidenced by the very few publications on CE using LSSP). In my opinion the adoption of the capture vaporizer that may lead to CE~1 for ambient particles (Xu et al., 2016; Hu et al., 2016; and work in preparation that is documented in relevant presentations at AMS users meetings http://cires1.colorado.edu/jimenez-group/wiki/index.php/AMSUsrMtgs) may be much more useful for the vast majority of AMS (and especially ACSM users). The authors may want to comment on the relative advantages and disadvantages of both methods, which is a topic of current high-interest for the AMS community.

Other issues

(18) P2 L36: it is unclear what the end of the sentence about matrix effects means. Has someone shown that there are no matrix effects? If so, reference(s) should be provided. Or is that an assumption? Or do the authors have unpublished data that shows this?

(19) P3 L90: this statement is very incorrect for m/z 57. It is well-known that m/z 57 is only dominated by $C_4H_9^+$ for pure HOA-type species, but in SOA its contribution is often smaller than for the main oxygenated ion at that m/z, $C_3H_5O^+$. See for example P25 of the Supp. Info. of Aiken et al. (2009), but this has been reported in many studies. Many types of SOA reported by Docherty et al. are very oxidized, and this statement will certainly be incorrect for those.

(20) P6 L178, this is not completely correct. Two particles may arrive so close in particle time-of-flight that they cannot be distinguished from the signal traces. Presumably the probability of this phenomenon is low, and should be quantifiable with the available data. This should be briefly mentioned and quantified.

(21) P7 L225, which AMS size distribution was used for alignment? (LS or MS)

(22) P9, L285, I would add "for ion formation" to "for ion extraction." E.g. a particle may end up in a location where the vapors do not overlap efficiently with the electron beam. In fact I would expect ion formation to be the dominant effect.

(23) P10, L315-319. Related data has recently been published by Hu et al. (2016). In their Fig. 14 these authors report that the lowest temperature at which monodisperse particles of a given species are detected in a narrow size distribution is proportional to the melting and boiling/decomposition temperatures of the species.

(24) P13 L439, I think this should be "thermal decomposition" (of neutrals) rather than

fragmentation (of ions).

(25) Fig 7, the fact that a-pinene SOA bounces similarly to ammonium sulfate is a useful result that could be highlighted. Even though Bahreini et al. (2005) and Docherty et al. (2013) have reported similar results, this is still lost on many researchers.

(26) P14 L478-479. Perhaps a better explanation is that the particles shrank due to evaporation, and were too small for the LSSP mode? Poulain et al. (2010) (Fig 7) and Huffman et al. (2009) (Fig 4) show that ∼20-30% of the mass of a-pinene SOA remains detectable by the AMS at 110C (and those numbers are known to be biased low due to the lack of detection of particles smaller than ∼50 nm by the AMS).

Also, the thermodenuder is not described in the experimental section or here, and no reference is provided. This needs to be documented in the manuscript, including whether the temperature profile was accurately measured and how (which is relevant for the comparisons with other studies), and what the residence time was. The above results are for Aerodyne-style TDs with typical residence times of 15 s at room temperature. If the residence time was much larger in the TD used here, perhaps that could also explain the difference between the studies.

(27) P15, L491-492, wording is confusing and I think incorrect. The SOA from the present study is similar to the least oxidized OF ALL THE SOAs that had CEm ∼0.2 in Docherty et al. But the text currently refers to the least oxidized SOA in the Docherty study, which had f44/f57 of 0.06 and not 5. The difference of a factor of x2.5 between this and the Docherty study would seem important and could use more discussion.

(28) The abstract mentions substantial variability in CEp, but does not discuss whether it is random, or whether it is explained by some experimental parameters. Reading the paper it would seem that it would seem that it is mostly experimental variation, but I suggest that this is directly addressed and clarified.

Wording etc.

(29) P1 L10: "appear within the chopper window" will not be understood by non-AMS practitioners

(30) P2 L40: "the another"

(31) P2 L52, should be "vacuum aerodynamic diameter" as there are other definitions of aerodynamic diameter that apply for other instruments.

(32) P3 L64, 65, 66, "ToF" and "time-of-flight", when used alone, are easy to confuse with the mass spectrometer. For this reason the AMS community uses "PToF" to refer to "particle time-of-flight" and associated chambers, data, software etc. (as opposed to ion time-of-flight).

(33) P4 L99, it would be clear if the denominator read "All particles detected by LS". Otherwise it is ambiguous and potentially confusing.

(34) P4 L123, solution should be plural

(35) P4 L125, presumably the authors mean milliCuries (mCi) and not milliCoulombs as written

(36) P5, L138, "flash vaporization" would not create particles but vapors, unless it is followed by nucleation and condensation of the vapors. A clearer name for the technique may be "evaporation-condensation" or something like that.

(37) P5, footnotes: neither of the web links given in the footnotes work.

(38) P9, L299. Given that PhD theses are harder to obtain and are not peer-reviewed (at least not in the same way as papers), I suggest that this reference is replaced (or supplemented) by the peer-reviewed paper by the same author and year (Alfarra et al., 2004), which contains the same information cited here. This may apply to other instances of the same reference elsewhere in the current paper.

(39) P10, L304, many RIE_org values have recently been published (Jimenez et al., 2016).

(40) Fig 7b, perhaps a log horizontal axis would work better to see the trend to 1 s?

(41) P10 L319, Saleh et al. is not yet published in AS&T. (As an aside, the data mentioned in comment #23 may be relevant to that paper).

(42) P10, L328-330. The notation for CE' and CE is confusing. I recommend using instead M and P superscripts for the MS and PToF modes, respectively

References

Aiken, A. C., et al.: Mexico City aerosol analysis during MILAGRO using high resolution aerosol mass spectrometry at the urban supersite (T0) – Part 1: Fine particle composition and organic source apportionment, Atmos. Chem. Phys., 9, 6633-6653, doi:10.5194/acp-9-6633-2009, 2009.

Alfarra, M.R., et al. Characterization of Urban and Regional Organic Aerosols In the Lower Fraser Valley Using Two Aerodyne Aerosol Mass Spectrometers, Atmospheric Environment, 38: 5745–5758, 2004.

Bahreini, R., et al. Measurements of Secondary Organic Aerosol (SOA) from oxidation of cycloalkenes, terpenes, and m-xylene using an Aerodyne Aerosol Mass Spectrometer. Environ. Sci. Technol., 39, 5674-5688, 2005.

Hu, W., et al.: Evaluation of the new capture vaporizer for Aerosol Mass Spectrometers (AMS) through laboratory studies of inorganic species, Atmos. Meas. Tech. Discuss., doi:10.5194/amt-2016-337, in review, 2016.

Huffman, J.A., et al. Chemically-Resolved Volatility Measurements of Organic Aerosol from Different Sources. Env. Sci. Technol., 43, 5351–5357, 2009.

Jimenez, J.L. et al.. Comment on "The effects of molecular weight and thermal decomposition on the sensitivity of a thermal desorption aerosol mass spectrometer." Aerosol Science and Technology, 50, i-xv, doi:10.1080/02786826.2016.1205728, 2016.

Li, X. et al. EXPERIMENTAL AND NUMERICAL STUDIES ON THE NORMAL IMPACT

OF MICROSPHERES WITH SURFACES. Journal of Aerosol Science, 30, 439-449, 1999.

Poulain, L., et al.: Towards closing the gap between hygroscopic growth and CCN activation for secondary organic aerosols – Part 3: Influence of the chemical composition on the hygroscopic properties and volatile fractions of aerosols, Atmos. Chem. Phys., 10, 3775-3785, doi:10.5194/acp-10-3775-2010, 2010.

Salcedo, D., et al.: Determination of particulate lead using aerosol mass spectrometry: MILAGRO/MCMA-2006 observations, Atmos. Chem. Phys., 10, 5371-5389, doi:10.5194/acp-10-5371-2010, 2010.

Shilling, J. E., et al.: Loading-dependent elemental composition of $\alpha$-pinene SOA particles, Atmos. Chem. Phys., 9, 771-782, doi:10.5194/acp-9-771-2009, 2009.

Xu, W., et al. Laboratory characterization of an aerosol chemical speciation monitor with PM2.5 measurement capability, Aerosol Sci. Tech., 1-15, 10.1080/02786826.2016.1241859, 2016.

---

## Author Comment (AC1) · 9 Jan 2017

RC1 Response - AMTD CE - amt-2016-271 We appreciate the reviewer's time to read and comment on our manuscript. The feedback was helpful, and the suggested changes have made the paper stronger. We have organized our responses to the reviews by using the same numbering as the initial review, and then putting our responses in blue.

\*\*\*\*\*\*\* \*\*\*\*\*\*\* \*\*\*\*\*\*\* \*\*\*\*\*\*\* \*\*\*\*\*\*\* \*\*\*\*\*\*\* \*\*\*\*\*\*\* \*\*\*\*\*\*\* \*\*\*\*\*\*\* \*\*\*\*\*\*\* \*\*\*\*\*\*\* \*\*\*\*\*\*\*
\*\*\*\*\*\*\* \*\*\*\*\*\*\*

[Figure]

1. "Line 125: Bequerel (Bq) is the SI unit for activity." -We have changed the abbreviation of milliCuries to the correct "mCi" instead of how it was written before (mC).

2. "Line 339: Please explain what you mean by 'sizzle'." -We removed this sentence. The previous sentence sums up what we meant by "sizzle."

3. "Figure 3: Please use two more distinct colours." -We changed the magenta trace to a teal one to make these two traces more distinct.

4. "Figure 5: Rather than normalised frequency per bin, it would be better to do this as a probability density function (taking bin width into account). The data could then be legitimately plotted as lines (rather than sticks), making the figure easier to read." -We considered this suggestion at first, but don't think it quite works as well as the sticks. Our reason is the following: the null category has a very sharp distribution compared to the prompt and delayed distributions. Additionally, the split y-axis is required to show the null distribution (which would make the use of lines fairly confusing). Lastly, we choose not to bin our data any finer because the distributions become so noisy as to not properly convey the information visually. Anyway, considering all of these things, lines would work fine for the prompt and delayed particles, but not the null, and so to keep everything uniform this is the best we feel we can do with this plot.

---

## Author Comment (AC2) · 9 Jan 2017

RC2 Response - AMTD CE - amt-2016-271

We would like to thank the reviewer for their time looking over our manuscript. This feedback really truly very helpful, both in it being thorough and thoughtful. Thank you very much for reading the paper. We have organized our responses to the reviews by using the same numbering as the initial review, and then putting our responses in blue.

\*\*\*\*\*\*\* \*\*\*\*\*\*\* \*\*\*\*\*\*\* \*\*\*\*\*\*\* \*\*\*\*\*\*\* \*\*\*\*\*\*\* \*\*\*\*\*\*\* \*\*\*\*\*\*\* \*\*\*\*\*\*\* \*\*\*\*\*\*\* \*\*\*\*\*\*\* \*\*\*\*\*\*\* \*\*\*\*\*\*\* \*\*\*\*\*\*\*

[Figure]

1. "A key problem is that the manuscript, as currently written, will be very confusing for most AMS users. The wording is quite specific and a clear connection to the way most users utilize the AMS is not given anywhere. My estimate is that 99% of published AMS figures use MS-mode data only, âĹij1% use PToF mode, and âĹij0.1% or less use LSSP mode. This may become even more skewed towards MS-mode data in the future as the number of ACSMs (that only have MS mode) surpasses and then greatly exceeds the number of AMSs in use. Thus it seems critical to clarify the implications explicitly. This could be done with a section such as "Implications for AMS and ACSM" users before the conclusions, or maybe a paragraph in the conclusions." ⁃We appreciate the fact that the LSSP module is more towards the 'specialized' end of the AMS user spectrum, and we appreciate the feedback that our message more applicable to those who use LSSP and also PToF than for those users that don't. We feel that that should be quite clear to anyone reading this article. However, this does not seem like a limitation from our perspective, but rather a simple aspect of what we have done. We do not agree that is a key problem, as stated in this comment. AMT is a "measurement techniques" journal. Sometimes very specific papers need to be published describing the nuts and bolts. This paper is also not about the ACSM so we elect not to speculate about the ACSM. We feel that by nature of being in this journal, with the title that we have written, the people interested in this subject will find this paper should they be interested.

2. "There is a small fraction of the signal for non-refractory species that is detected over even longer timescales than the MS mode open / closed alternation, and thus contributes to slower variations in the AMS background signals ("MS closed"), as documented in e.g. Huffman et al. (ACP 2009, cited in the manuscript). For semi- refractory species, such very slow signals can be larger and even dominant, but even then they can be used to extract particle concentrations (see e.g. Salcedo et al. 2010). Did the authors examine the variations in the background of the mass spectrometer (MS closed) during their experiments? It most likely contains information that is very relevant to this paper, in particular signals from the evaporation of bounced particles.

This should be analyzed and discussed in the revised paper."  We aren't exactly sure what the reviewer is looking for here. Indeed, there is mass detected in the closed spectrum, albeit much less than is detected in the open spectrum. We even see an increase in the ratio of closed:open mass for increasing TD temperatures. These things indicate that the SOA we have made contains low-volatility compounds, which is not a new finding. How exactly this should be made relevant to the current manuscript is not clear. There are numerous indications that LS and MS modes do not detect things in the same way. Even this is not a new finding, see Cross et al, 2009, but we have further demonstrated it in this paper e.g. Figure 7b. We do not clearly see what is gained in discussing the closed spectrum in addition to what is already presented. Throughout the manuscript there are mentions of the idea that mass detected in MS mode is done so on long timescales (relative to the chopper cycle), that particles counted as "null" in LS mode do provide some chemical information in MS mode, etc. We don't mean to discount this comment, but it is not clear what the reviewer is interested in here.

3. "An important limitation of the results of this study is that the SOA was formed at extremely high concentrations (1500 ug/m3). Those concentrations are not atmospherically-relevant, and the composition of the SOA formed cannot be expected to be the same as for much lower, atmospherically-relevant concentrations. For example typical concentrations at Hyytiala, Finland (a location where a-pinene SOA is important) are âĹij1 ug/m3, a factor of 1000 smaller than in this study. Shilling et al. (2009) showed that the composition of a-pinene + O3 SOA changed very strongly between 0.5 and 140 ug m-3, with very large changes below 30 ug m-3. The results of this study, while important in terms of characterizing the AMS detection system, should not be recommended for application for ambient particles in terpene-dominated environments. The fact that the SOA was generated at very high concentrations, and thus that it is not clear whether the results will be applicable to ambient SOA from this precursor and oxidant, needs to be stated prominently in the abstract and conclusions of the paper"  We have made explicit that these high concentrations are not atmospherically-relevant for SOA, and have cited Shilling et al 2009. The following sentence has been

inserted: "The composition of SOA is loading-dependent, as demonstrated by Shilling et al 2009, and so preparing SOA at this necessarily-high concentration is a potential limitation of this work, as is often the case for laboratory studies of SOA systems." (P5L L136)

4."The need to use extremely high concentrations would appear to be a critical limitation of this technique. It is important that the author discuss what the lowest concentrations that could possibly be studied with the LSSP method are, as well as possible methods to allow performing similar experiments at lower SOA concentrations. Did the authors try to use small seeds? E.g. using 10 nm seeds from (for example) an electrospray aerosol source), that would then be coated by a much larger amount of SOA. Presumably the CE would be dominated by the SOA coating, and insensitive to the initial seed. The authors are creative experimentalists and may have better suggestions, and it would be useful to document those in the paper" ‐As per our response to the previous comment (#3), we have tried to draw explicit attention to the fact that these experiments were conducted at high concentrations, the reasons for conducting our experiments in this way, and that, as always, there may be limitations when considering the atmospheric relevance of such laboratory results in application to e.g. an ambient dataset. Making SOA particles big enough to study, given the experimental methods we chose, required high concentrations. We elected against using seeds due to experimental ease, potential confounding issues regarding morphology, and wanting to look at the CE of 'pure' SOA. Other methods for future studies may be better, though there will always be a trade-off in balancing the LS size threshold requirement, the ratio of volume between seed and coating (so as to only be 'studying' the CE of the coating), particle morphology, and the required concentrations to produce a 'big-enough' coating thickness. These considerations are beyond the scope of the work we presented, though we hope that other researchers may improve upon these experiments and we can learn from them and further refine our own methods in the future. Our assessment is that those reading this work will be familiar with these issues, but we have tried to draw attention to the issue of high concentrations with our response to the previous

comment.

5."Fig. 5 and P9 L271. Apparently a very short chopper time (5 ms) was used for these experiments. This is unfortunate, as the AMS can typically be operated at >10 ms, which would have provided better data. Similarly the mass spec. pulsing frequency was too long (30 us), and a shorter frequency would have allowed recording single particle events in more detail. Both items should be recommended in this paper to future experimenters wishing to use this technique, so that improved data can be acquired" ⁃We recognize the limitations of these two issues in this dataset, as both the chopper timing and pulser frequency, as identified by the comment, were not optimal. While we did point both of these issues out in the original text, we have tried to further highlight this. We have added the following to identify these limitations, though we do not feel as if they are deal-breakers for conveying the important points of this paper. On P10 L298, the following was added to emphasize this point about the short chopper cycle used here: "It should be noted that the length of the chopper cycle used in these experiments was not optimal, and that a longer cycle would allow us to see the most-delayed particles. We recommend a long chopper cycle for ambient measurements and/or any experiments where delayed particles may be expected." To address the point about the pulser period, the following sentence has been added to P12 L385: "It should be noted that the pulser period used in this work is a limitation; a faster pulser period should be used for future similar work, as it would allow for proper quantification of the event length."

6."P10, L301-307. I would argue that this is an extremely important point. I.e. the confusion of many users about the potential variability in CE, and the large effort needed to estimate it accurately, has obscured any trends in RIE that might be present on the laboratory and field data. Methods such as in the current paper where CE is quantified allow investigating RIE, which is a timely topic. It seems to me that this should be mentioned more prominently, e.g. in the abstract and conclusions." ⁃The first line in the abstract does already contain the phrase "effective ionization efficiency," and the

first two paragraphs of section 3.3 discuss this in detail, so the discussion isn't exactly hidden. However, we have added another line at the end of the abstract to try to make this (investigating RIE) more prominent: "By measuring the mean ions per particle produced for monodisperse particles as a function of signal delay time, we can separately determine CEp and CEm and thus more accurately measure the relative ionization efficiency (compared to ammonium nitrate) of different particle types."

7. "P11, L336-342. An alternative hypothesis (that appears much more likely than the 2 mentioned here) is that a particle bounces from the vaporizer without any evaporation, and then lands into a colder surface in the ionizer. The AMS ionizer cage is heated to âĹij250C by the electron emission filament, and thus NO prompt evaporation at all should be expected for most species when impacting that surface. Given the geometry of the AMS ionizer, most bounced particles will impact the AMS ionizer cage. If they exit the cage region, they will end up on surfaces which are much farther and much colder (âĹij30C). This hypothesis can explain why substantial more particle mass is detected in MS mode than in PToF mode, as the colder temperatures (âĹij250C) result in far slower evaporation. Can the authors incorporate this hypothesis into their discussion, even if these particles may not be detected at all in PToF mode? ⁃The 'alternative hypothesis' presented in this comment is exactly what we are saying—that SOA particles, at least the ones visible in LSSP and PToF data, are bouncing from the vaporizer and landing on other surfaces in the ionization region. However, we still do very clearly see 'prompt' (meaning fast) evaporation from our delayed particles. In retrospect, it was an oversight that we did not explore colder vaporizer temperatures in our SOA experiments, as we could test directly whether we would see any prompt evaporation in LSSP data at e.g. 250 celsius. However, we did not run these tests. Nonetheless, this is an empirical study and we "see what we see," so to speak. For the particles we do observe, there is no change in the peak width for 2.5 ms of delay time, though the average ions per particle drops by a factor of 3-4 in the first 1 ms of delay time (Figure ). That is simply observed. There is no evidence that the peaks are getting wider in the 2.5 ms span, though in that time period the particles have bounced 20 cm total distance (assuming they retain their initial velocity)— it seems implausible that particles have not bounced a fair bit. Further, there is absolutely no sign of periodicity in the total signal for the 2 ms period, and so we conclude that the average distance between bounces is more or less random — there is no evidence that the particles are traveling away and then back or just away some specific distance to a unique surface that is hot enough to vaporize them. Thus, they are either traveling away, bouncing off of lots of different sites, and returning to the hot vaporizer or they are traveling away to sites that (in the first 2.5 ms) still evaporate them quickly. We have tried to be much more clear about null particles, which do contribute mass in MS mode, but are not visible whatsoever in LSSP or PToF modes. These null particles are a missing part of the story, when it comes to LSSP and PToF data, which we have tried to be clear about.

8. "P11, L344, this result can be influenced by "threshold bias". I.e. some particles may start to evaporate more slowly per the previous comment, but not be detected at all in by the ion threshold used in single particle mode. This appears obvious from the distribution of signals of the delayed particles in Fig. 5, which would have kept increasing in frequency towards lower signals than the threshold used here. Maybe only those particles that impact the ion cage very close to the filaments, in an area that is likely much hotter than 250C, may lead to the sharp delayed particles. Overall while this discussion is valuable, it is incomplete. A physical schematic (cartoon) of the different possibilities for particle trajectories and associated signals would be useful for readers of the papers and for future discussions (e.g. at AMS Users Meetings and in future papers). This should include particles completely missed due to threshold bias under slow evaporation. I strongly encourage the authors to add it here, and include the additional possibilities mentioned in the previous paragraphs (even if the present data do not allow characterizing all of them). Similarly, on P11, L375: this hypothesis is not proven with the current data, since other particles may have been completely missed due to threshold bias." ⁃Following from the previous comment and our accompanying answer, we have made much more clear that LS and PToF data only see particles that evaporate within the chopper window. However, there are

enough delayed particles with substantial mass to say that the sharp peaks seen for delayed SOA particles are not an artifact of threshold bias. We have addressed the mass threshold (which is the dividing line between null and prompt/delayed particles) in various places throughout the manuscript. For example, we explicitly refer to sub-threshold particles in LS still providing mass that is meaningful for bulk operation of the AMS. Importantly, this results in an apples-to-oranges comparison for LS and MS calculations of collection efficiency. See P9 L283 for a mention of this, as well as P10 L304. As for the temperature profile within the vaporizer and ionizer region, these numbers are news to us, and we don't quite feel comfortable making conclusions based on them. Additionally, the vaporizer itself is a long tube that runs approximately the length of the ionization cage, all of which should be the same temperature as the particle-striking surface of the vaporizer.

9. "P12, L388, and L395-397. This text and axis label should be "upper limit of distance bounced" as particles will lose energy and reduce their speed upon collision. The con-clusion on L397 about many bounces being common is not really supported. E.g. if the particles lose 90% of their speed upon the first impact, then they may just evapo-rate on the 2nd impact, with a distance that the authors would calculate as 10 cm, but would only really be 1 cm. There is some data on velocity changes upon impact for much larger particles (e.g. Li et al., 1999). I did not find results for submicron particles on a quick search, but they are probably out there. One would expect that submicron particles will lose a much larger fraction of their speed, due to the far larger ratio of adhesion forces to inertia, compared to the data in Li et al. The authors should do a more thorough search and use the results to inform this discussion" ⁃While we did state that this estimation of distance bounced is under the assumption of elastic colli-sions, we have further emphasized this by adding references to work done on velocity changes upon particle bounce, as suggested, and being very explicit that our distance bounced calculation is an upper limit estimate. We have also amended the axis label and figure caption for Figure 6 as suggested. We hope these changes illustrate that this is just a broad, back-of-the-envelope estimation of the very upper limit of bounce

distance, and could indeed be lower depending on the nature of the collision (which likely depends on many factors e.g. particle composition, angle of impact(s), etc.). We hope these changes address the concerns raised in the above comment.

10. "P12 L401-402: this conclusion could be made if adhesion/adsorption forces did not change between the two temperatures. But for example Fig. 11 of Hu et al. (2016) shows that chloride detection in the standard vaporizer becomes slower when going from 600 to 835 C. This effect is not observed for other species, indicating that it depends on the specific chemistry of the system being studied. This seems to be the case here as well, see next comment. So I don't think that the conclusion about where this tail comes from is as obvious as the authors state. Note that the ionizer surfaces will be somewhat hotter with a hotter vaporizer, due to radiation heat transfer which is efficient at those temperatures."  At this stage in the manuscript we have established that: 1) the SOA in our study has a long tail in its PToF distribution (Figure 2), 2) the tail is attributable to delayed particles (Figure 4), and 3) that these delayed SOA particles have narrow evaporation profiles (Figure 8a), unlike e.g. ammonium nitrate when striking the vaporizer at 200C (Figure 8b). Based on the analysis of LS data, it is clear where the tail comes from, insofar as whether it is prompt or delayed particles. We present Figure 9a simply to say that the PToF tail can not be explained by the evaporation rate of SOA, which would increase with a hotter vaporizer. Aside from the chloride example, where detection is slower at higher temperatures (which, as far as we can tell, does not have an explanation in the Hu manuscript, though we may have missed it?), all examples we have seen have followed the trend of faster detection at higher vaporizer temperatures.

11. "L403-405. This statement is incorrect. It is clear from the data in Fig. 9a (of this paper) that SOA detection has a LONGER tail at 800C. Although the signal is noisy, a fit or smoothing of the data (as I did just by eye) clearly reveals this feature. Also only the front baseline region should be used to provide a zero level for the data ("DC marker region"), which will also increase the difference. At present the average of the

front region is clearly negative." -The statement is not incorrect. As we say in the text, we do not see the tail go away or diminish with the increase in vaporizer temperature, and Figure 9a illustrates this. Whether the tail increases is another matter. The signal to noise for this PToF distribution is quite high, and our intention was for this plot to be a qualitative assessment on the hypothesis that increased vaporizer temperatures would decrease the PToF tail, which we feel it serves to answer. Further tests would be needed to quantitatively assess the degree to which the tail potentially increases, as the reviewer does point out, but we feel that the illustration of whether or not there still is a tail at 800C addresses the point we were trying to make in a qualitative sense. We have corrected the issue highlighted by the front PToF region being negative. We had applied the DC markers, but failed to update the figure version in the document; this was a mistake and one we appreciate being corrected.

12."P12 L404-406, here it is unclear whether the signal plotted in Fig 9c is exclusive to a-pinene SOA. If the m/z 43 signal arises from both squalane and SOA (as expected from the NIST spectrum of squalene http://webbook.nist.gov/cgi/inchi?ID=C111013&Mask=200#Mass-Spec, and noting that the AMS spectrum will have substantially higher m/z 43 than the NIST one), it is possible that the signal shown in this figure is dominated by squalane, and thus that the conclusion reached here is incorrect. The authors have experience identifying signals dominated by one or another species/sources in binary mixtures, and that method should be applied to improve this graph and hopefully clarify whether the conclusion is correct." ⁃We apologize for this mistake—the squalane used here is fully-deuterated (d62) squalane, which was stated in the Methods (along with a list of the deuterated squalane ions seen in the AMS) section but referred plainly to "squalane" here and in other parts of the text. All squalane used in this project was d62-squalane. The reviewer's comment above would certainly be true for SOA coated with non-deuterium labelled squalane but is not an issue for the isotopically-labeled version used here. We have updated the text so that mentions of squalane are properly identified as "d62- squalane" to avoid any confusion.

13. "P13 L 409-420. This conclusion can only be made if the arrival times (initial onset of the signal) are shown for NH4NO3, which is not currently done. Fig 9b seems to show a secondary mode that could be due to delayed particles. The author should investigate and document this." -It seems that the confusion is arising due to the secondary mode of particles, which in this experiment arises from multiply charged particles. We have identified this in the updated text, so as not to confuse the reader. What should be clear from Figure 9b, however, is that the primary mode of particles significantly broadens when the vaporizer temperature is lowered. We hypothesize that this is due to slower evaporation, and feel that Figure 8b is very consistent with that hypothesis where we show the single-particle average arrival profile being significantly longer at the low vaporizer temperature. This result is similar to what Docherty et al, 2015 show as well.

14."P13 L 434-436 and Fig 10: the NH4+ fragment is shown for ammonium sul- fate. This is typically not detected in AMS spectra (e.g. Table 1 of Hu et al., 2016, which is derived from high-resolution data). Presumably the authors are analyzing unit- resolution data and mislabeled the ion. This fragment is almost certainly HÂËĞn2O+, a major ion observed for (NH4)2SO4. More H2O is expected to arise arise from the decomposition of H2SO4, a process more important for the prompt particles, as Fig. 10b shows. It is important to fix this as it could otherwise cause a lot of confusion in the AMS community. (Alternatively, if the authors had high-resolution evidence of NH4+ detection at such a large fraction as shown in the figure, it would be big news)." ⁃This label was erroneous, and has been changed. Indeed, the fragment at m/z 18 is H2O+, as identified in the high-resolution MS, and not NH4+ as the original text stated.

15. "P14 L 466-467: How can LSSP separate prompt and delayed particles of different sizes, when sampling a polydisperse aerosol? It seems (from e.g. Fig. 4) that this would be extremely difficult, unless all particles were prompt, which is not the case here. This is important since the polydisperse data in the blue experiment seem to

show an increase with size, while the (much more trustable, in my opinion) monodisperse data do not."  We base our size-measurement (Dva) for single-particles on the time-of-flight measured between the chopper and the laser. As this paper demonstrates, the Chopper-MS time-of-flight measurement of Dva can be subject to significant error if there is any delay in vaporization. The chopper-LS Dva measurement is not affected by particle bounce, and so we can compare the measured size across the different single- particle categories (prompt, delayed, null, etc.). We have attempted to be more clear in the revised manuscript by emphasizing that the size for the polydisperse experiments refers to the chopper-LS time-of-flight, and that the size-selected mobility diameters were converted to Dva according to eqn. 5. See P15 L508 of the updated text, where we added the following text: "...estimated from the time-of-flight between the chopper and the laser. Importantly, this measure of size is unaffected by any vaporization delays, and can be compared across LS particle categories (e.g. prompt, delayed, null)."

16. "P2 L35: Jimenez et al does not state that RIE_org is assumed to be the same for all organic species. It does actually provide some cross-section data that could allow grouping hydrocarbons and oxygenates separately. The assumption of the same RIE_org for all organics in the field was introduced later based on experimental results at Aerodyne, and to my knowledge Canagaratna et al. (Mass Spec. Rev. 2007) may be the first publication of that assumption. To my knowledge it has never been published that all organics in the lab should be analyzed with RIE = 1.4, rather than calibrated for each experiment (especially given wide variability in CE of organics, as documented by e.g. Bahreini et al. (2005) and Docherty et al. (2013). Although that has been a common misinterpretation. See Jimenez et al. (2016) for further discussion of this topic."  We have cleaned up the language here, and correctly referred to Canagaratna, 2007. The original text read "assumed to be roughly similar," so it was not a blanket statement that all organic species have the exact same RIE. Nonetheless, the sentence has been changed to address this issue.

[Figure]

17. "P16 L522-524. Returning to issue #1 and the implications for most AMS/ACSM users. I would argue that for the vast majority of AMS users, CE IS a serious limitation. While the information obtained is undoubtedly of high interest, very few AMS groups have the technical skill, equipment, and time to perform and interpret analyses such as the ones in this paper (as evidenced by the very few publications on CE using LSSP). In my opinion the adoption of the capture vaporizer that may lead to CEâĹij1 for ambient particles (Xu et al., 2016; Hu et al., 2016; and work in preparation that is documented in relevant presentations at AMS users meetings http://cires1.colorado.edu/jimenez-group/wiki/index.php/AMSUsrMtgs) may be much more useful for the vast majority of AMS (and especially ACSM users). The authors may want to comment on the relative advantages and disadvantages of both methods, which is a topic of current high- interest for the AMS community." ⁃This paper is a rather nuts and bolts paper about issues related to bouncing particles with the standard vaporizer platform. This paper is not about the capture vaporizer (CV), and we elect not to discuss the CV as it is not within the scope of this paper. The CV papers that are coming out can speak much more effectively to these issues than we are able to, and we look forward to reading them as they come out.

18. "P2 L36: it is unclear what the end of the sentence about matrix effects means. Has someone shown that there are no matrix effects? If so, reference(s) should be provided. Or is that an assumption? Or do the authors have unpublished data that shows this? " ⁃Because the AMS uses thermal vaporization followed by EI, there is no reason to expect matrix effects in the EI and we are aware of no published evidence that there ARE matrix effects. We have reworded this sentence.

19. "P3 L90: this statement is very incorrect for m/z 57. It is well-known that m/z 57 is only dominated by C4H9+ for pure HOA-type species, but in SOA its contribution is often smaller than for the main oxygenated ion at that m/z, C3H5O+. See for example P25 of the Supp. Info. of Aiken et al. (2009), but this has been reported in many studies. Many types of SOA reported by Docherty et al. are very oxidized, and

this statement will certainly be incorrect for those." ⁄The reviewer correctly points out an error here. We have changed that sentence to the following: "Docherty et al. (2013) report an inverse relationship between CEm for chamber-generated SOA and the f44/f57 ratio (where m/z is comprised of $CO_2^+$ , while m/z 57 is comprised of the less oxidized marker fragments $C_4H_9^+$ and $C_3H_5O^+$, and $f_i$ is the fraction of ($m/z_i$) to the total organic signal)." (P4 L93)

20. "P6 L178, this is not completely correct. Two particles may arrive so close in particle time-of-flight that they cannot be distinguished from the signal traces. Presumably the probability of this phenomenon is low, and should be quantifiable with the available data. This should be briefly mentioned and quantified." ⁄This is true, and a valid concern given the high number concentrations of mono-disperse particles used in some of these experiments, as the chance of coincident particles (meaning multiple particles per chopper cycle) increases with increasing number concentration. And, coincidence is not an issue, as it can be identified except in the case where two particles have exactly the same time-of-flight between the chopper and laser, as the reviewer points out. We agree, in this case, which is rare, coincidence can't be identified. However, the likelihood of this happening, even at very high number concentrations, is quite low. Consider the following: using the number distribution of particles from one of our mono-disperse experiments and the time-resolution of the light-scattering trace recorded from the PMT in the AMS software, we estimate that the algorithm used to identify coincident particles would fail 3% of the time that two particles entered a single chopper cycle. If the mono disperse particle size distribution were a delta function, it would fail 100% of the time that two particles showed up in the same chopper cycle. For experiments where our particle number concentrations were highest, coincidence as determined by the AMS algorithm was 13%. Assuming 13% is approximately equal to the actual number of chopper cycles where coincidence truly occurred, we can say that for 3% of those cycles we failed to identify coincidence, and counted two very-similarly sized particles as one. This is only 0.4% is the percent of all chopper cycles where coincidence would have occurred but would be unidentifiable using the algorithm

employed by SQUIRREL. We have updated the text to provide this estimate, which is low enough not to affect the results stated in the paper. The paragraph identified in this comment has been amended to reflect this calculation and better address identifying coincident particles

21. "P7 L225, which AMS size distribution was used for alignment? (LS or MS)" –The SMPS size distribution was aligned with the AMS PToF size distribution. We have changed the language to make this more clear: "For this example experiment, where 370 nm SOA particles were size-selected using a DMA, shown in Figure 2, we estimate the density to be 1.1 g/cm3 from aligning the mode diameters of the SMPS-calculated mass distribution with that from the AMS mass distribution measured in PToF mode." (P8 L250)

22."P9, L285, I would add "for ion formation" to "for ion extraction." E.g. a particle may end up in a location where the vapors do not overlap efficiently with the electron beam. In fact I would expect ion formation to be the dominant effect." –We have changed the text to "for ion formation and extraction." (P10 L315)

23. "P10, L315-319. Related data has recently been published by Hu et al. (2016). In their Fig. 14 these authors report that the lowest temperature at which monodisperse particles of a given species are detected in a narrow size distribution is proportional to the melting and boiling/decomposition temperatures of the species." –This is interesting, though the arrival time of squalane vs. ammonium nitrate in our work would go against the trend of reported by Hu for salts, as squalane has a lower boiling point than ammonium nitrate. As stated in the paper, other work by Saleh et al and Murphy indicate volatility and/or molecular weight as affecting the timescale of detection in the AMS, even at the standard vaporizer temperature. Whatever the reason, we hope the biggest takeaway from Figure 7a and this paragraph in question is that ammonium nitrate and squalane look more similar to eachother, despite their differences, from SOA and AS, both of which are known to bounce (which we show as well). The Hu paper demonstrates, at least for 3 different inorganic salt species, a different concept:

they are showing the transition point where size distributions widen. We are showing that prompt particles arrive at slightly different times for two species that have not gone past the 'transition point' presented by Hu et al. Regardless, we have refined the language in this paragraph to make more clear that we don't know for sure what causes the slightly slower arrival of individual particle signals for d62- squalane compared to ammonium nitrate, but that we are speculating on possibilities that are consistent with the aforementioned works being cited. A more thorough analysis of all of these factors would be interesting, but is outside the scope of this paper.

24."P13 L439, I think this should be "thermal decomposition" (of neutrals) rather than fragmentation (of ions)."  This change was made.

25."Fig 7, the fact that a-pinene SOA bounces similarly to ammonium sulfate is a useful result that could be highlighted. Even though Bahreini et al. (2005) and Docherty et al. (2013) have reported similar results, this is still lost on many researchers."  We completely agree, and have tried to highlight this. See P11 L356 for statements regarding this.

26. "P14 L478-479. Perhaps a better explanation is that the particles shrank due to evaporation, and were too small for the LSSP mode? Poulain et al. (2010) (Fig 7) and Huffman et al. (2009) (Fig 4) show that âĹij20-30% of the mass of a-pinene SOA remains detectable by the AMS at 110C (and those numbers are known to be biased low due to the lack of detection of particles smaller than âĹij50 nm by the AMS)."  This is an important potential point that hypothetically could bias our measurements, though we have done our analysis such that this should not be a concern. One of the nice aspects of the LSSP module is that there are two independent ways to detect a particle and its size: light scattering and mass detection. P16 L522 addresses the issue brought up by this comment: we can limit our analysis to particles above a certain size range, in order to not confound CE changes with simply small particles that do not have enough detectable mass. This sentence addresses this issue: "The CEp values in Figure 11 are calculated for particles with 200 nm > Dva >300nm to isolate the effects

of volatility and/or oxidation state on CEp, instead of measuring smaller particles less likely to provide enough detectable mass above the threshold." The other issue raised in this comment is that the thermodenuder used needs to be sufficiently detailed. A paragraph on the thermodenuder has been added to the end of section.

27. "P15, L491-492, wording is confusing and I think incorrect. The SOA from the present study is similar to the least oxidized OF ALL THE SOAs that had CEm âĹij0.2 in Docherty et al. But the text currently refers to the least oxidized SOA in the Docherty study, which had f44/f57 of 0.06 and not 5. The difference of a factor of x2.5 between this and the Docherty study would seem important and could use more discussion." ⁃The confusion here arises, we think, from the fact that some of the data presented in the Docherty study (the least oxidized data points in Figure 6) do not correspond to SOA at all, but are other pure standards (e.g. dioctyl sebacate). We have tried to make this statement less confusing by changing this sentence. We feel that the change communicates more clearly the message that our results broadly agree both in oxidation state and CEm.

28. "The abstract mentions substantial variability in CEp, but does not discuss whether it is random, or whether it is explained by some experimental parameters. Reading the paper it would seem that it would seem that it is mostly experimental variation, but I suggest that this is directly addressed and clarified." ⁃This is correct. We were unable to identify and explain the cause for experiment-to-experiment variability in CEp. We have added language to both the abstract and conclusions to further emphasize this. See following on P17 L558: "CEm for SOA across all experiments was 0.49($\pm$0.07S.D.) while CEp was 0.30($\pm$0.04S.D.), though we were unable to explain the reason for the variability in CEp between different experiments."

29. "P1 L10: "appear within the chopper window" will not be understood by non-AMS practitioners" ⁃We have changed this sentence to the following: "SOA particles measured in construction the AMS mass distribution..." has replaced "SOA particles that appear within the chopper window..."

30. "P2 L40: "the another" ⁃We fixed this sentence, removed "the."

31. "P2 L52, should be "vacuum aerodynamic diameter" as there are other definitions of aerodynamic diameter that apply for other instruments." ⁃We fixed this sentence, added "vacuum."

32. "P3 L64, 65, 66, "ToF" and "time-of-flight", when used alone, are easy to confuse with the mass spectrometer. For this reason the AMS community uses "PToF" to refer to "particle time-of-flight" and associated chambers, data, software etc. (as opposed to ion time-of-flight)." ⁃We fixed this sentence, changed "time-of-flight" to "Particle time-of-flight" to avoid any ambiguity.

33. "P4 L99, it would be clear if the denominator read "All particles detected by LS". Otherwise it is ambiguous and potentially confusing." ⁃We changed the denominator to "All particles detected by LS."

34. "P4 L123, solution should be plural." ⁃This change was made.

35. "P4 L125, presumably the authors mean milliCuries (mCi) and not milliCoulombs as written." ⁃Fixed.

36. "P5, L138, "flash vaporization" would not create particles but vapors, unless it is followed by nucleation and condensation of the vapors. A clearer name for the technique may be "evaporation-condensation" or something like that." ⁃We felt that the sentences following the use of "flash vaporization" make clear that squalene is vaporized, and that those vapors, upon cooling, form particles. However, we've tried to make this point more clear and replaced this with the above suggestion.

37. "P5, footnotes: neither of the web links given in the footnotes work." ⁃We've updated the links to reflect the new web addresses.

38. "P9, L299. Given that PhD theses are harder to obtain and are not peer-reviewed (at least not in the same way as papers), I suggest that this reference is replaced (or supplemented) by the peer-reviewed paper by the same author and year (Alfarra et

al., 2004), which contains the same information cited here. This may apply to other instances of the same reference elsewhere in the current paper."  The reference provided in this comment does not contain the information we are referring to, unlike the thesis does. Other papers (e.g. Alfarra, 2006; Docherty, 2013) have referred to this thesis, which is available on the Jimenez group AMS library page.

39. "P10, L304, many RIE_org values have recently been published (Jimenez et al., 2016)."  We have added the Jimenez et al reference here.

40. "Fig 7b, perhaps a log horizontal axis would work better to see the trend to 1 s?"  The current figure, in our minds, communicates well the balance of the different information presented by this data: the different timescales of particle arrival and the different amount of particle bounce between the different aerosol types investigated.

41. "P10 L319, Saleh et al. is not yet published in AS&T. (As an aside, the data men- tioned in comment #23 may be relevant to that paper)."  We will update this reference; that paper will be published by the time this paper would hypothetically be in print.

42."P10, L328-330. The notation for CE' and CE is confusing. I recommend using instead M and P superscripts for the MS and PToF modes, respectively."  We are already using the p subscript for "particle" collection efficiency, vs. the m subscript for "mass." Given how limited our use throughout the manuscript is of CE' (which is defined in this paragraph, and is the collection efficiency defined by comparing the PToF integrated mass vs the density-corrected SMPS mass) vs CE, we are going to elect to keep these the same.